# Understanding the lithium–sulfur battery redox reactions via operando confocal Raman microscopy

Shuangyan Lang [1], Seung-Ho Yu [2], Xinran Feng[1], Mihail R. Krumov[1] & Héctor D. Abruña [1] ✉

The complex interplay and only partial understanding of the multi-step phase transitions and reaction kinetics of redox processes in lithium–sulfur batteries are the main stumbling blocks that hinder the advancement and broad deployment of this electrochemical energy storage system. To better understand these aspects, here we report *operando* confocal Raman microscopy measurements to investigate the reaction kinetics of Li–S redox processes and provide mechanistic insights into polysulfide generation/evolution and sulfur deposition. *Operando* visualization and quantification of the reactants and intermediates enabled the characterization of potential-dependent rates during Li–S redox and the linking of the electronic conductivity of the sulfur-based electrode and concentrations of polysulfides to the cell performance. We also report the visualization of the interfacial evolution and diffusion processes of different polysulfides that demonstrate stepwise discharge and parallel recharge mechanisms during cell operation. These results provide fundamental insights into the mechanisms and kinetics of Li–S redox reactions.

Lithium–sulfur (Li–S) batteries represent one of the most promising candidates of next-generation energy storage technologies, due to their high energy density, natural abundance of sulfur, and low environmental impact. Li–S redox involves multi-step chemical and phase transformations between solid sulfur, liquid polysulfides, and solid lithium sulfide ($Li_2S$), that give rise to unique challenges in Li–S batteries. A critical issue is the formation and "shuttle effects" of soluble intermediate polysulfides during battery operation. Polysulfides bridge the solid insulating sulfur and lithium sulfide ($Li_2S$), and partially mitigate the high resistance and sluggish electrochemical reactions of pure solid sulfur. However, transport of polysulfides away from the cathode via diffusion results in loss of active material, causing severe capacity fade and passivation on both electrodes[1–3].

With the intent of addressing the polysulfide shuttle phenomenon, substantial efforts have been devoted to regulating/electrocatalyzing polysulfide redox pathways and restricting polysulfide diffusion by developments in the design of the cathode, electrolyte, and interlayer[4–8]. Despite numerous reports on approaches to mediate polysulfides, the underlying mechanistic details of the fundamental Li–S redox kinetics and interfacial polysulfide evolution processes remain elusive. State-of-the-art in situ/*operando* characterization techniques, including synchrotron-based methods[9–12], electron microscopy (EM)[13,14], atomic force microscopy (AFM)[15,16], nuclear magnetic resonance (NMR) spectroscopy[17–21], Raman spectroscopy[22–25], and ultraviolet–visible (UV–vis) spectroscopy[26–28], are helping establish the knowledge base for Li–S reaction mechanisms[29–31]. X-ray based methods have been extensively employed to distinguish/identify different sulfur species and unravel their transformation pathways during Li–S redox processes[9,32–34]. *Operando* transmission X-ray microscopy and in situ X-ray fluorescence microscopy have enabled mapping the evolution of sulfur and related polysulfide intermediates[35–37]. Using X-ray absorption spectroscopy, Wang et al. quantified the average chain

[1]Department of Chemistry and Chemical Biology, Cornell University, Ithaca, NY 14853, USA. [2]Department of Chemical and Biological Engineering, Korea University, Seoul 02841, Republic of Korea. ✉e-mail: hda1@cornell.edu

length of polysulfides upon discharge, presenting the first estimate of relative rate constants for polysulfide transformations[38]. A thorough understanding of the complex Li−S redox requires the combination of high-resolution imaging, simultaneous identification of the different (poly)sulfide species, rapid and multi-site detection, and quantitative analysis. For most of the characterization tools, the limited resolution in spatial distribution and/or the lack of clear distinction of different polysulfides restrict their accurate characterization and comprehensive diagnosis of the reaction mechanism and kinetics of the Li−S redox processes.

Raman measurements are well suited to examine the reaction pathways during/of Li−S redox processes due to the strong Raman intensities of sulfur and polysulfides, and their convenience for carrying out *operando* studies. Using Raman methodologies, various reaction mechanisms/pathways have been proposed (Supplementary Table 1), which illustrate the key roles of the solvent[22–24], additives[25,39] and cathode materials[40] on the evolution of polysulfides and overall battery performance. Wu et al. directly applied a first-order kinetics analysis to fit the sulfur reduction data[25]. The simple combination of the multi-step redox processes using high overpotentials and the lack of analysis focusing on polysulfides, led them to conclude that short-chain $S_3^{·-}$ is formed directly from sulfur reduction, which does not agree with the generally accepted views about the solid-liquid-solid transformations and the initial formation of long-chain polysulfides during sulfur reduction[3,11]. In fact, fundamental insights into many crucial issues of Li-S redox have not been established, including the reaction order of different sub-steps, the potential and concentration dependency of reaction rates, the nucleation and growth processes of Li$_2$S, and transformation mechanism and kinetics between/among different polysulfides. This is partly because of the complexity of the multistep Li-S redox reactions, involving solid-liquid-solid phase transformations and the diffusion of soluble polysulfides. In addition, it is quite difficult to simultaneously track, identify, and quantify the different (poly)sulfide species in real-time during the discharge and charge processes. Confocal Raman microscopy enables the simultaneous identification of sulfur and polysulfides, and the high-resolution imaging of their spatial distribution, are very promising and powerful for the systematic and quantitative analysis of the Li−S redox mechanism and kinetics[30,41].

Here, we explore the use of *operando* confocal Raman microscopy to investigate and quantify Li−S redox processes, based on both potentiostatic and galvanostatic measurements, achieving a detailed characterization of their reaction pathways and kinetics. By visualizing the potential-dependent reactants and intermediates, quantifying the changes in their intensities, and comparing among various classical models, we demonstrate the first-order reaction kinetics of sulfur reduction and polysulfide redox processes. The conductivity dependency of sulfur reduction and the concentration dependency of polysulfides have also been investigated/characterized. We further elucidated the nucleation and growth mechanisms of electrodeposited Li$_2$S and S upon redox. Fundamental insights into the correlations between/among the overpotential, the shape of the current-time transients, the morphologies of the electrodeposited Li$_2$S, and the corresponding capacities are provided. During galvanostatic discharge and charge processes, the Raman images captured at the cathode surface, and deep into the electrolyte, clearly illustrate the spatial distribution and differences in intensity changes of polysulfides at different states of discharge/charge, suggesting a stepwise reduction process, but a parallel oxidation mechanism during recharge. Using *operando* confocal Raman microscopy, our work provides compelling experimental evidence of sulfur reduction kinetics and mechanistic insights that extend the knowledge of Li-S redox processes. Moreover, it stimulates further exploration of Raman imaging to investigate multiple complex reaction processes.

## Results

### Potentiostatic reduction of sulfur clusters

The reduction of elemental sulfur involves the complex multi-step generation and evolution of polysulfides during the discharge process. In a typical electrolyte of 1.0 M lithium bis(trifluoromethanesulfonyl) imide (LiTFSI) in 1,3-dioxolane (DOL)/1,2-dimethoxyethane (DME) (1:1), the reduction pathways have been widely investigated by various experimental methods and theoretical calculations[22,26,42]. The sulfur reduction involves the ring-opening and chain-shortening processes as described in the expression below:

$$x/8\,S_8 + 2e^- \rightarrow \text{long-chain } S_x^{2-}\,(x=6-8) \xrightarrow{+2e^-} \text{intermediate } S_x^{2-}\,(x=3-5) \xrightarrow{+2e^-} \text{short-chain } S_x^{2-}\,(x=1-2) \quad (1)$$

Additional specific reactions, including some typical reduction substeps and possible disproportionation and conproportionation reactions are presented in Supplementary Table 2. A representative cyclic voltammetric (CV) profile is shown in Supplementary Fig. 1a. In the initial reduction process, two cathodic peaks at 2.30 and 1.97 V are observed, corresponding, respectively, to the reduction of sulfur to soluble polysulfides and their further reduction forming Li$_2$S[43,44]. On the reverse scan, the anodic peaks at 2.36 and 2.42 V are ascribed to the reversible oxidation back to elemental sulfur[45]. In an effort to understand the dynamic evolution of the equilibrium voltages, galvanostatic intermittent titration technique (GITT)[46] measurements were carried out during the discharge process (Supplementary Fig. 1b). After each equilibration, the open-circuit voltages ($E_{\text{relax}}$) were extracted from the raw data (Supplementary Fig. 1c). The average values of $E_{\text{relax}}$ were found to be 2.35 ± 0.02 and 2.15 ± 0.01 V for the upper and lower plateaus, which are linked to the reduction reaction of sulfur and intermediate polysulfides, respectively (Supplementary Fig. 1d).

Figure 1a presents a schematic illustration of the experimental setup for *operando* confocal Raman microscopy to precisely visualize the sulfur clusters and sulfides, based on their characteristic Raman signals, as well as follow their time evolution. Changes of the potential-dependent concentrations of reactants and intermediates during the Li−S redox processes can be quantified, providing essential insights into kinetic parameters and underlying mechanistic details. Since the reaction rate is a strong function of potential, we employed chronoamperometric measurements, where a constant potential is applied, to investigate the reduction of sulfur. Initially, a potential of 2.30 V (vs. Li$^+$/Li), was selected representing an overpotential ($\eta$) of about 50 mV ($E_{\text{relax}}$ = 2.35 V). The current was recorded as a function of time (*I*–*t* curve), as shown in Fig. 1b. We observed sulfur clusters on the carbon fiber current collector (schematic: inset of Fig. 1a, optical image: Supplementary Fig. 2a) and examined their evolution at 2.30 V. The corresponding optical images are shown in Supplementary Fig. 2. Supplementary Fig. 3 presents the full spectrum of the sulfur sample, showing multiple intense and well-defined peaks (at 87, 152, 220 and 475 cm$^{-1}$) as well as some smaller/shoulder peaks (at 245 and 436 cm$^{-1}$) which are typical of S$_8$. With increasing time, the characteristic Raman peaks of sulfur at 152, 220, and 475 cm$^{-1}$ decreased, followed by the emergence of new peaks at 405 cm$^{-1}$ and 453 cm$^{-1}$ (Fig. 1c). The peaks at 405 cm$^{-1}$ and 453 cm$^{-1}$ were employed to illustrate the evolution of the long-chain Li$_2$S$_x$ ($x$ = 6–8) and intermediate-chain Li$_2$S$_x$ ($x$ = 3–5) polysulfides, respectively, during potentiostatic and galvalnostatic experiments[22,24,29,47]. Supplementary Fig. 4 presents the spectra of sulfur and the polysulfides at the same intensities. We then selected the bands at 220 cm$^{-1}$ (Fig. 1c–h, red) and 453 cm$^{-1}$ (Fig. 1c–h, yellow) as representative of sulfur and long-chain polysulfides, respectively, to map the evolution of sulfur to polysulfides. The sulfur clusters dissolved from the edges to the center (Fig. 1d–h and Supplementary Movie 1), consistent with the three-phase interfacial reaction between

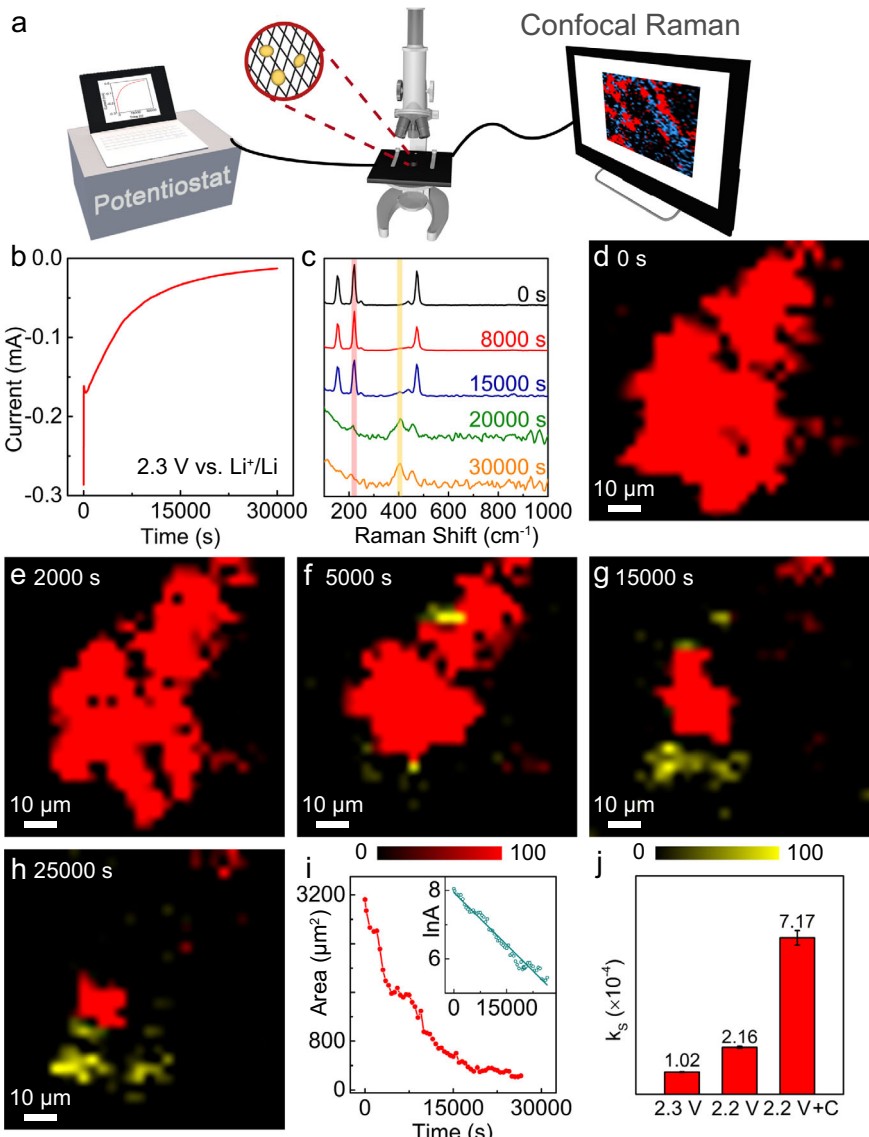

**Fig. 1 | Reduction of sulfur clusters at 2.30 V vs. Li⁺/Li. a** Schematic illustration of the *operando* confocal Raman microscopy experimental setup for probing the mechanism and kinetics of Li–S redox processes. **b** Chronoamperometric current-time transient of sulfur reduction at 2.30 V (vs. Li⁺/Li). *Operando* Raman (**c**) spectra and (**d**–**h**) mapping images of the sulfur electrode during reduction. The red and yellow colors in (**d**–**h**) represent sulfur and long-chain polysulfides, respectively, where the color contrasts remained consistent for quantification. **i** Plot of the area changes of sulfur clusters with time. Inset: Linear fitting of the logarithm of the area with time according to first-order kinetics, $R^2 = 0.979$. **j** Values of $k_S$ acquired at 2.30 V, 2.20 V, and 2.20 V using the cathode with an additional carbon interlayer. The error bars represent the standard deviation of the linear fits.

the sulfur, carbon cathode, and electrolyte. The polysulfides, which can be visualized at the interface after 5000 s (Fig. 1f), gradually diffused to the side areas (Fig. 1g, h). Upon continuation of the discharge process, changes in/around the sulfur clusters slowed down, with some residual polysulfides being trapped (Fig. 1g, h).

In an effort to understand/establish the relationship between the active surface area of sulfur clusters and the reaction rate, we plotted the area change of the sulfur clusters over time (Fig. 1i). (N.B.: This approach implicitly assumes that the area serves as a proxy for concentration.) The rate of the sulfur reduction reaction was expressed in terms of the following rate expression:

$$-\frac{dA}{dt} = k_S A^n \qquad (2)$$

where $k_S$ represents the apparent rate constant for the reduction rate of sulfur per active site, $A$ is the active surface area of sulfur

which, in this work, is represented by the projected area from Raman mapping, and n is the reaction order with respect to sulfur. The concentration of Li⁺ is in excess at the interface. The reaction order, n, can be estimated by fitting the data to various rate expressions with typical models of zero, first, second, and third-order kinetics (Fig. 1i and Supplementary Fig. 5). A plot of ln$A$ versus time exhibited a linear decrease (inset of Fig. 1i), indicating pseudo-first-order reaction kinetics for the reduction of sulfur. The rate constant at 2.30 V was calculated to be $1.02 \times 10^{-4} \pm 0.02 \times 10^{-4} \text{ s}^{-1}$, where the deviation is largely due to the difference between the actual active surface area and the projected area. For example, the overlap of the active reaction areas that are perpendicular to the observation direction are difficult to separate from the projected area, which would lead to a smaller value of $k_S$.

To further verify the potential-dependent kinetics of sulfur reduction, potentiostatic experiments were also performed at an applied potential of 2.20 V (vs. Li⁺/Li) with a larger overpotential

($\eta$ = 150 mV, $E_{relax}$ = 2.35 V). The obtained results are shown in Supplementary Fig. 6 and Supplementary Movie 2. At 2.20 V, both Raman spectra and mapping images indicate a decrease of the pristine sulfur clusters and the formation of polysulfides, as the reduction proceeded. The linearity of a plot of lnA vs. time confirms that the sulfur reduction reaction follows a first-order rate expression. The value of $k_S$ was calculated to be $2.16 \times 10^{-4} \pm 0.05 \times 10^{-4}$ s$^{-1}$ at 2.20 V, which is about twice the value at 2.30 V. The results indicate the accelerated rate of sulfur reduction with larger overpotentials.

The electronically insulating properties of sulfur, hinder the full utilization of sulfur cathodes and thus impact their practical application. Incorporating an electron conductive matrix into the sulfur cathode composite has been widely employed to maximize the contact area and facilitate sulfur conversion[4,48,49]. In an effort to understand the influence of the electronic conductivity of the sulfur cathode composite on the reduction reaction of sulfur, an additional carbon layer was further deposited onto the sulfur cathode. *Operando* confocal Raman experiments were performed at 2.20 V (vs. Li$^+$/Li) using the sulfur cathode with a carbon interlayer (Supplementary Fig. 7a–e). Based on the quantification of the sulfur area changes with time, we concluded that under these conditions the reduction of sulfur also follows a first-order rate process, and that the reaction rate significantly increased, when compared to that without the carbon interlayer. The linear fits and detailed analysis are shown in Supplementary Fig. 7f–i. Figure 1j provides the calculated rate constants at 2.30 V ($1.02 \times 10^{-4} \pm 0.02 \times 10^{-4}$ s$^{-1}$), 2.20 V ($2.16 \times 10^{-4} \pm 0.05 \times 10^{-4}$ s$^{-1}$), in the absence pf the carbon layer and at 2.20 V using a cathode with an additional carbon interlayer ($7.17 \times 10^{-4} \pm 0.33 \times 10^{-4}$ s$^{-1}$). It shows that the rate constant, with the carbon interlayer, increased by a factor of 3.5 relative to the one without it, indicating that the enhancement of the electronic conductivity plays a significant role in increasing the density of active sites and in accelerating the reaction rates of sulfur redox. This emphasizes the importance of micro-/nano- encapsulation of sulfur with a conductive matrix for achieving high percent utilization, which, in turn, could enhance overall device performance.

**Visualizing the evolution of polysulfides**

From the above discussion, the potentiostatic reduction processes of sulfur, and its reaction kinetics, have been explored focusing mainly on the first plateau region (2.35–2.2 V) during the discharge process. The intermediate polysulfides are formed on the second plateau (2.15–2.0 V) during the discharge process, where they can be reduced further to short-chain polysulfides and eventually to Li$_2$S[42,45]. In an effort to investigate these processes in more detail, we performed potentiostatic reduction studies using Li$_2$S$_4$, as a prototypical/representative intermediate polysulfide as the catholyte. CV and GITT experiments were also performed and results are presented in Supplementary Fig. 8. The cathodic peak at 2.04 V, corresponding to the reduction of intermediate polysulfides to short-chain polysulfides, is evident from the CV (Supplementary Fig. 8a), as well as the discharge plateau, with a value of $E_{relax}$ of $2.16 \pm 0.02$ V, which was verified by GITT (Supplementary Fig. 8b–e). These results are fully consistent with those in which we used sulfur as the cathode, indicating and validating our approach/protocol to use Li–Li$_2$S$_4$ cells to investigate the reaction of the polysulfides during the overall sulfur reduction processes.

A full Raman spectrum of the catholyte, trapped in the carbon electrode, is shown in Supplementary Fig. 9. The peak at 400 cm$^{-1}$, ascribed to S$_x^{2-}$, $x$ = 6–8, was observed because these polysulfide anions can be interconverted in solution due to their close Gibbs free energies[50]. The disproportionation and conproportionation reactions of the polysulfides can easily take place and maintain dynamic equilibria in solution during the entire discharge and charge processes. As shown in Supplementary Fig. 10, the signals remained stable for extended time periods, indicating that the equilibria of the dis- and con- proportionation reactions are established rapidly and that our detection would not be affected by the complex side reactions.

Cells using 1.0 M Li$_2$S$_4$ electrolyte were then employed to characterize the evolution of polysulfides. Figure 2 shows their reduction processes at 2.0 V (vs. Li$^+$/Li) ($\eta$ = 160 mV, $E_{relax}$ = 2.16 V). The current-time transient in Fig. 2a presents the reduction and phase transition processes from soluble polysulfides to short-chain polysulfides[51]. Raman peaks at 205, 400, and 453 cm$^{-1}$ decreased in intensity and eventually disappeared with increasing time (Fig. 2b). The region around 453 cm$^{-1}$ was selected for quantification, revealing the reduction of intermediate polysulfides (Fig. 2b–f, blue). As time progressed, the blue areas diminished in size and color intensity, indicating the gradual decrease of their concentrations (Fig. 2c–f and Supplementary Movie 3). The optical images in Supplementary Fig. 11 also show a decrease in color intensity. The reduction rate of the polysulfides can be expressed by the following differential equation:

$$\frac{dC}{dt} = k_{ps}C^n \qquad (3)$$

where $k_{ps}$ is the apparent rate constant for the reduction rate of polysulfides per active site, $C$ is their concentration normalized to the surface area from Raman mapping (for example, the area at open circuit in 1.0 M Li$_2$S$_4$ cell serves as a proxy for a concentration of 1.0 M), and n is the reaction order with respect to the polysulfides. Determination of the reaction order was performed using kinetic models as shown in Fig. 2g, h and Supplementary Fig. 12. The logarithm of the concentration was linear with time (Fig. 2h), indicating the first-order kinetics of their reduction. The slope of the curve ($k_{ps}$), calculated in 1.0 M Li$_2$S$_4$ electrolyte was found to be $1.60 \times 10^{-3} \pm 0.09 \times 10^{-3}$ s$^{-1}$, where the deviation could be, and likely is, affected by the disproportionation reactions of the different polysulfides. The Raman peak of long-chain S$_x^{2-}$, $x$ = 6–8 polysulfides (at 400 cm$^{-1}$) was also analyzed. It showed a rapid decrease and then remained at low concentrations after the initial 500 s (Supplementary Fig. 13), indicating its faster reduction rate at the relatively larger overpotential ($\eta$ = 350 mV) at 2.0 V.

Potentiostatic reduction experiments were performed at 2.0 V with different initial concentrations of Li$_2$S$_4$. The reduction rates slowed down at the lower concentrations of 0.75 and 0.5 M Li$_2$S$_4$, as shown in Fig. 2g. The corresponding plots in Fig. 2h exhibited a linear relationship between the logarithm of the concentration and time, confirming the first-order dependence of the reaction rate on the concentration. Detailed results and analysis in 0.75 and 0.5 M Li$_2$S$_4$ electrolytes are shown in Supplementary Figs. 14 and 15. According to Eq. (3), the first-order rate constants ($k_{ps}$) were calculated to be $1.76 \times 10^{-3} \pm 0.14 \times 10^{-3}$ s$^{-1}$ and $1.88 \times 10^{-3} \pm 0.15 \times 10^{-3}$ s$^{-1}$ in 0.75 and 0.5 M respectively, which shows good consistency with the value in 1.0 M Li$_2$S$_4$ electrolyte (Fig. 2i). These results indicate that *operando* confocal Raman microscopy is a powerful approach to directly visualize and precisely quantify the evolution of polysulfides, in combination with chronoamperometric measurements, expanding our understanding of the reaction kinetics of Li–S redox processes.

Given the previous observations one may ask, how would the insoluble Li$_2$S nucleate and grow during the potentiostatic reduction of polysulfides? Previous efforts have been devoted to investigating various factors that could influence the electrodeposition of Li$_2$S, such as the solvent[52], the cathode host material[53], and the electrolyte/sulfur ratio[54]. However, the effects of one of the most fundamental factors of Li$_2$S electrodeposition, mainly the driving force, have not been clearly articled nor understood up to date. In an effort to address these issues, we performed potentiostatic reduction experiments at different applied potentials using Li–Li$_2$S$_4$ cells. As shown in Supplementary Figs. 16–19 and Supplementary Note 1, the current-time transients, I–t curves, performed at higher overpotentials showed increased peak currents and nucleation rates, which could be related to a change in

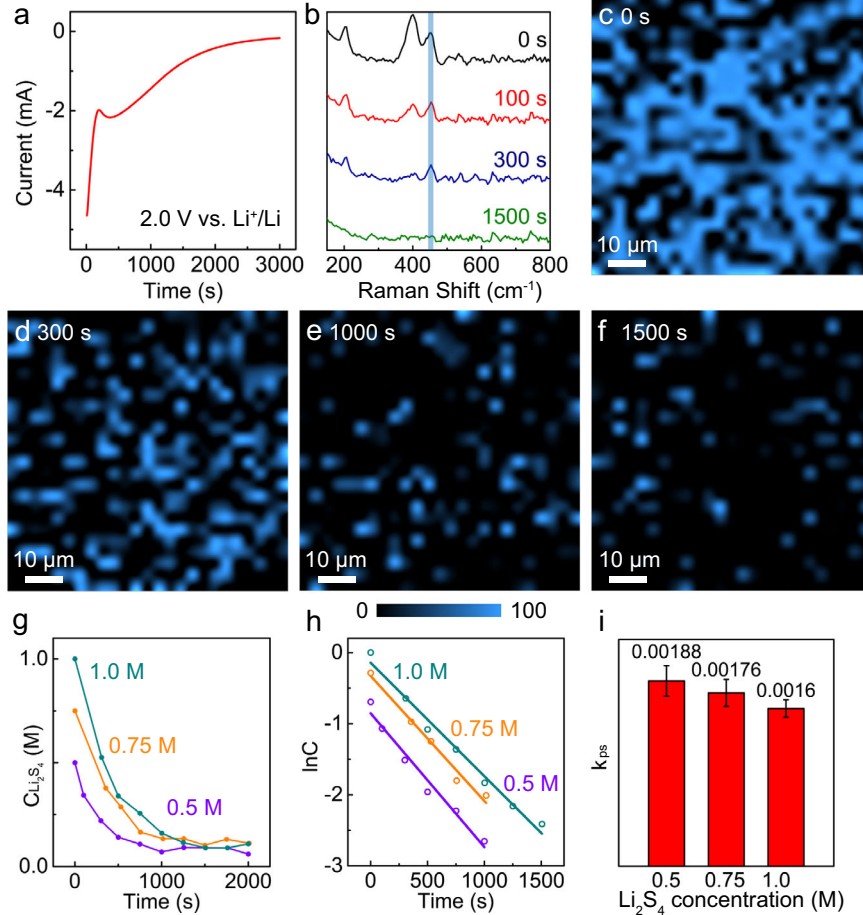

**Fig. 2 | Reduction of polysulfides at 2.0 V vs. Li⁺/Li. a** Chronoamperometric current-time transient of polysulfide reduction at 2.0 V (vs. Li⁺/Li). *Operando* Raman (**b**) spectra and **c**–**f** mapping images of the cathode during reduction in 1.0 M $Li_2S_4$ electrolyte. The blue color in **c**–**f** represents the region at 400 cm⁻¹. The color contrast remains consistent for quantification. **g** Plots of the concentration changes in 1.0, 0.75, and 0.5 M $Li_2S_4$ electrolytes with time. **h** Linear fitting of the logarithm of $Li_2S_4$ concentration evolution with time according to first-order kinetics, $R^2 = 0.982$, 0.979, 0.968 for 1.0, 0.75, and 0.5 M electrolytes, respectively. **i** Values of $k_{ps}$ extracted from the fits. The error bars represent the standard deviation of the linear fits.

the nucleation and growth mechanism of $Li_2S$ by fitting to the Bewick–Fleischmann–Thirsk (BFT)[55] and Scharifker–Hills (SH)[56] models (Supplementary Table 3).

## Sulfur formation during polysulfide oxidation

The re-oxidation processes of polysulfides were investigated by a combination of current-time transient analysis and *operando* Raman visualization. Supplementary Fig. 20 presents the GITT results of the subsequent recharge process of Li–S cells. The $E_{relax}$ of polysulfide oxidation was 2.38 ± 0.01 V, representing the equilibrium voltage of the liquid-solid transition from polysulfides to elemental sulfur. A potential of 2.40 V (vs. Li⁺/Li) was selected for a detailed analysis of the potentiostatic oxidation processes. Figure 3a, b present the current-time transient at 2.40 V and the corresponding dimensionless analysis. It should be noted that even at lower overpotentials ($\eta$ of about 70 mV), the nucleation rate of sulfur is much higher when compared to that of $Li_2S$. This could be attributed, at least in part, to the higher binding energy of carbon and nonpolar sulfur, as a computational study previously reported[57]. The higher binding energy would indicate a greater tendency of sulfur to undergo electrodeposition, exhibiting a higher nucleation rate.

*Operando* Raman experiments were also performed in an effort to track, in real-time, the evolution of polysulfides and sulfur. As shown in Fig. 3c, the Raman peaks of the polysulfides decreased, followed by an increase of the peaks from sulfur at 1200 s. The regions at 220 cm⁻¹

(red) and 453 cm⁻¹ (blue) were selected for mapping to visualize the interfacial evolution (Fig. 3d–h and Supplementary Movie 4). After 200 s, the homogeneity of the catholyte was disrupted by the formation of circular sulfur structures, indicating the initial appearance of elemental sulfur (Fig. 3d, e). (NB: The straight line of sulfur on the top right of Fig. 3e is due to the presence of a carbon fiber.) This is evident when comparing the Raman mapping with the optical images (Supplementary Fig. 21). More sulfur clusters grew over time, as shown in Fig. 3f, g, and accumulated along the carbon fibers. The increase of sulfur clusters slowed down after 4900 s, in line with the decreasing current (Fig. 3h), indicating the gradual completion of the oxidation reaction and reactant depletion.

Similar to the reduction processes of sulfur and polysulfides, the oxidation of polysulfides and regeneration of sulfur can be expressed by the rate expressions:

$$\frac{dA_{ps}}{dt} = k_{ps}A_{ps}{}^n \tag{4}$$

$$\frac{dA_S}{dt} = K_s A s^n \tag{5}$$

The meaning of the symbols is the same as those in Eqs. (2) and (3), while $k_{ps}$ and $k_S$ in Eqs. (4) and (5) represent the rate constants for the oxidation of the polysulfides and regeneration of sulfur, respectively.

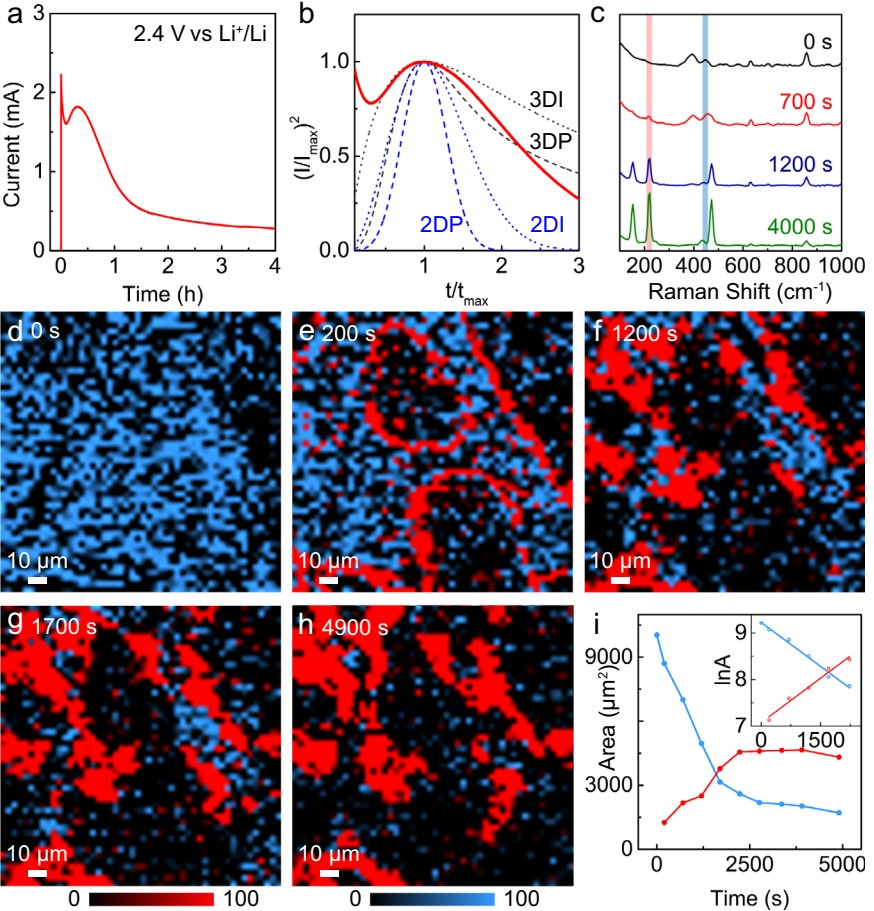

**Fig. 3 | Oxidation of polysulfides at 2.40 V vs. Li⁺/Li. a** Chronoamperometric current-time transient of polysulfide oxidation at 2.40 V (vs. Li⁺/Li). **b** Dimensionless plot of (**a**) according to SH and BFT models. *Operando* Raman (**c**) spectra and (**d**–**h**) mapping images during oxidation. Blue: intermediate polysulfides, Red: elemental sulfur, the color contrasts remain consistent in (**d**–**h**) for quantification. **i** Plots of the changes of the polysulfides (blue, decrease) and sulfur (red, increase) with time. Inset: Linear fit of the logarithm of the area changes with time according to first-order kinetics, $R^2 = 0.984$ and 0.972 for the polysulfides and sulfur, respectively.

Since the $Li_2S_4$ catholytes are homogeneous solutions, the changes of the areas from Raman mapping are assumed to be proportional to the changes in concentrations. For comparison with that of sulfur, we directly employed the area change in this case. As shown in the area change curves and the corresponding linear fits (Fig. 3i), both the evolution of the polysulfide oxidation and sulfur regeneration followed first-order kinetics. It is reasonable to expect that the concentration of the polysulfides and electronic conductivity (contact area) of sulfur would affect the rates of their reaction processes as well. Correspondingly, values for $k_{ps}$ and $k_S$ were calculated to be $6.29 \times 10^{-4} \pm 0.36 \times 10^{-4}$ s⁻¹ and $6.36 \times 10^{-4} \pm 0.54 \times 10^{-4}$ s⁻¹. The consistency of these two values indicates that side reactions are not significant in our case and illustrates the accuracy and reliability of the method employed.

Subsequently, larger overpotentials were applied for investigating sulfur nucleation and growth processes. Supplementary Fig. 22 presents the current-time transients at 2.45 V and 2.50 V (vs. Li⁺/Li), respectively. It is interesting to note that the transient at 2.45 V deviates from traditional models. There were split peaks, and even multiple peaks at 2.50 V. Similar phenomena of multi peaks have been previously reported in the study of metallic surface passivation at larger overpotentials, which is due to multilayer deposition[58]. As stated above, we ascribe our results to deviations from traditional 2 or 3 D nucleation/growth models[59,60]. Sulfur clusters were then observed via Raman mapping. As shown in Supplementary Fig. 23, the sizes of the sulfur clusters after 1 h of electrodeposition at 2.45 V and 2.50 V were smaller than those at 2.40 V, which is related to the limited growth after the non-classical nucleation process. This is also consistent with the fast decrease of the currents after the peaks.

## Li–S redox processes during galvanostatic cycling

In an effort to understand the Li–S redox processes under cell operation, we performed galvanostatic (constant current) discharge and charge studies on Li–S coin cells, employing *operando* confocal Raman microscopy to carry out the real-time monitoring of the evolution of active materials at the cathode. A sulfur electrode, prepared by coating a slurry of sulfur and polyvinylidene fluoride (8:2) onto carbon paper, was used as the cathode, Li foil as the anode, and 1.0 M LiTFSI in DOL/DME (1:1) as the supporting electrolyte. The discharge profile of the Li–S cell during the first cycle at 0.02 C (1 C = 1672 mAh g⁻¹) is shown in Fig. 4a. Two distinct plateaus are evident at around 2.35 and 2.1 V, corresponding, respectively, to the reductions of elemental S, and intermediate polysulfides, which is consistent with our CV and GITT results; *vide-supra*. The shortening of the second plateau at around 2.1 V could be due, at least in part, to the formation of insulating $Li_2S$, that blocked the further discharge processes at the interface.

Figure 4b presents the *operando* Raman spectra collected at different depths of discharge (DODs). Sulfur peaks at 150 cm⁻¹, 220 cm⁻¹, and 475 cm⁻¹ were detected on the pristine sulfur electrode, which subsequently decreased and disappeared completely at about 2.5% DOD. Meanwhile, the peak of long-chain polysulfides ($S_x^{2-}$, $x = 6$–8) at 400 cm⁻¹ appeared at 1.5% DOD. The intensity of this peak increased

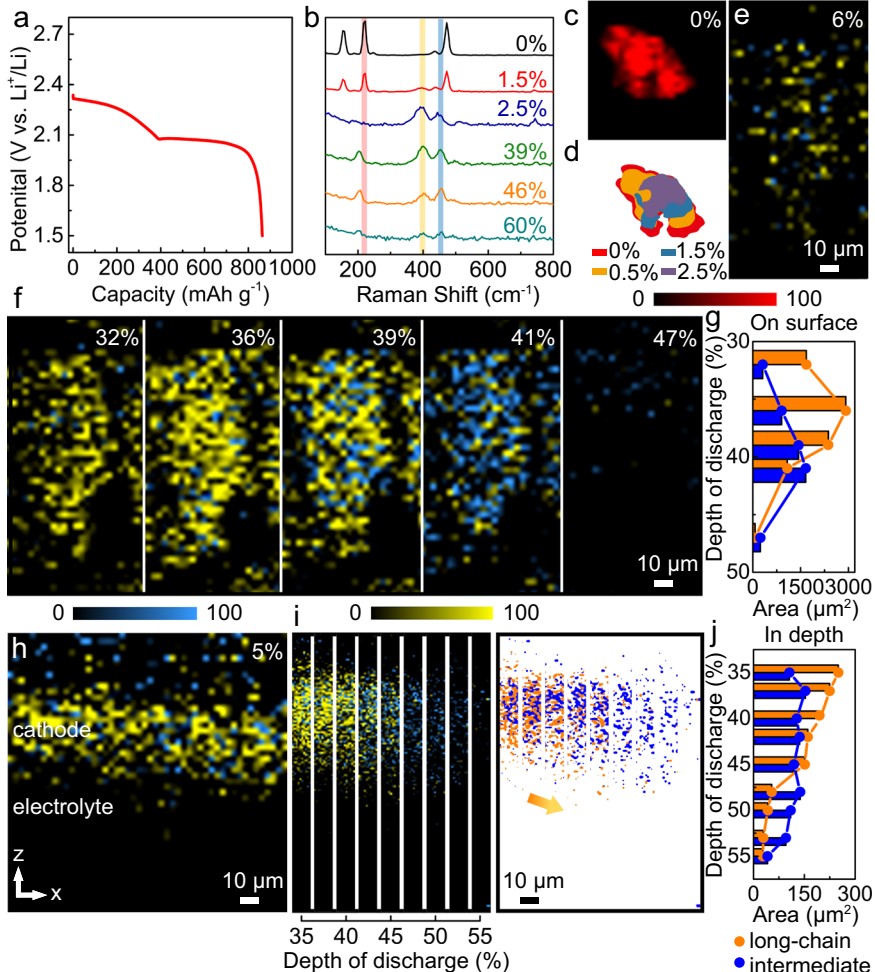

**Fig. 4 | *Operando* confocal Raman investigation of Li−S redox processes during the initial galvanostatic discharge process. a** Voltage profile, *operando* Raman (**b**) spectra and **c**−**j** mapping results of sulfur and polysulfides during discharge at a rate of 0.02 C (1 C = 1672 mAh g⁻¹). The red, yellow, and blue colors in mapping images represent sulfur, polysulfide $S_x^{2-}$, $x = 6−8$, and 3−5, respectively. The color contrasts remained consistent for comparison. Different DODs are indicated with %. **d** area changes of the pristine sulfur clusters at different DODs. **f** On surface and **i** in depth (Z-direction) mapping images of the polysulfide evolution during the discharge process. **g** and **j** Respectively, provide the quantified data on surface and in depth (Z-direction), extracted from **f** and **i**. The orange and blue colors represent long-chain and intermediate polysulfides, respectively. To the right of (**i**), for clear visualization, color mapping of cross-sectional polysulfide evolution was extracted from the Raman mapping images in the left. **h** In depth (Z-direction) mapping that reveals the cross-sectional distribution of the polysulfides at 5% DOD.

and remained at high values until 39% DOD (around the end of the upper plateau). The peak of intermediate polysulfides ($S_x^{2-}$, $x = 3−5$) at 453 cm⁻¹ can be clearly observed at 2.5% DOD. It increased in intensity and asymptotically reached a maximum at about 46% DOD, as the lower plateau was approached. Note that the peak of the long-chain polysulfides decreased during 39−46% DOD, while the peak of intermediate polysulfides slowly increased, indicating the reduction and transition from long-chain to intermediate-chain polysulfides. The peaks of both long-chain and intermediate-chain polysulfides decreased during the latter part of the lower plateau, which is attributed to the further reduction of polysulfides into insoluble Li₂S as well as to their diffusion processes.

The morphological evolution of the sulfur electrode was mapped via confocal Raman microscopy (Fig. 4c−f). The regions centered at 220 cm⁻¹ (red), 400 cm⁻¹ (yellow), and 453 cm⁻¹ (blue) were selected, as representing sulfur, long-chain $S_x^{2-}$, $x = 6−8$ and intermediate $S_x^{2-}$, $x = 3−5$ polysulfides, respectively. These changes, as a function of DOD, provide more detailed and direct evidence of the multi-step sulfur reduction processes. The initial sulfur clusters observed in Fig. 4c dissolved from the edges to the center during the discharge process (Fig. 4d). The pristine sulfur clusters disappeared completely at around 6% DOD, along with the appearance of the soluble polysulfides at the

interfaces (Fig. 4e). Supplementary Movie 5 and Fig. 24 present the mapping images and the corresponding quantifications of the polysulfides during the entire discharge and (re)charge processes. In Fig. 4f the 32−47% DOD region was selected to carefully analyze the reduction process of the polysulfides. The area changes are presented in Fig. 4g, where the orange and blue colors represent long-chain and intermediate polysulfides, respectively. Non-synchronous changes in the intensities of the long-chain and shorter chain polysulfides were observed over 35−40% DOD. The decrease of the long-chain and the increase of the intermediate-chain polysulfides indicate the transformation from long-chain to short-chain polysulfides, which is key to the evolution of short-chain polysulfides during discharge and consistent with a stepwise process. Polysulfides with different lengths co-exist in solution during different states of discharge as a result of the close Gibbs free energies and the con- and dis- proportionation reactions among them. The stepwise evolution mechanism proposed here is based on the changing trends of different polysulfides as characterized by the relative intensities and their changes. The longer-than-expected region of the long-chain polysulfides could be due, at least in part, to their slower diffusion processes relative to the short-chain ones and the sluggish solid-liquid reduction from sulfur, when compared to the liquid-liquid transformations between/among polysulfides.

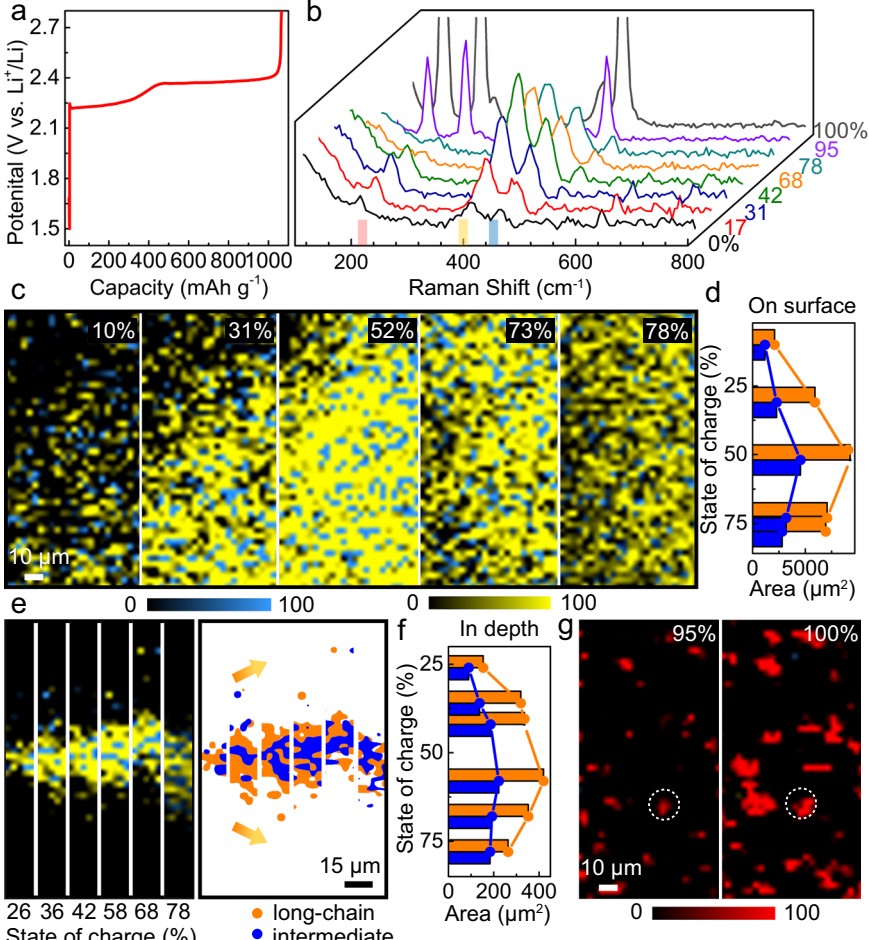

**Fig. 5 | *Operando* confocal Raman investigation of Li–S redox processes during the galvanostatic charge process in the initial cycle. a** Voltage profile, *operando* Raman (**b**) spectra, **c** on surface mapping and **e** in depth (Z-direction) mapping images of polysulfides during different SOCs at a rate of 0.02 C (1 C = 1672 mAh g⁻¹). Right side of **e**: color mapping changes of the in depth (Z-direction) mapping images. **d, f** quantified data of polysulfide evolution and diffusion on surface and in depth (Z-direction), extracted from **c, e**. The orange and blue colors represent long-chain and intermediate polysulfides, respectively. **g** Sulfur regeneration near the end of the charge process. The red, yellow and blue colors in mapping images represent sulfur, polysulfide $S_x^{2-}$, x = 6–8, and 3–5, respectively, where the color contrasts remained consistent for comparison. Different SOCs are indicated with %.

We also analyzed the diffusion processes (from cathode surface to electrolyte) of the polysulfides via confocal Raman microscopy. Figure 4h presents the cross-sectional mapping near the sulfur electrode at 5% DOD. Polysulfides were formed from the reduction of the pristine sulfur clusters and remained around the cathode. We focused on a 10 μm wide region and mapped the polysulfide evolution processes during 35–55% DOD (Fig. 4i). The reconstructed color images (orange: long-chain, blue: intermediate polysulfides) are presented in the right of Fig. 4i. The front edges of the polysulfide gradually stretch into the electrolyte, driven by the concentration gradient, which further decreases the signals at the edges. The disappearance/fading of these signals can be related to various reasons, including their further dilution, disproportionation reactions, and diffusion to the regions that were not under observation. The corresponding area changes are presented in Fig. 4j. The long-chain polysulfides continued to decrease during 35–55% DOD, while the intermediate ones remained at high values and slowly decreased at 48% DOD. Despite numerous reports in recent years, the mechanism of sulfur redox processes remains unsettled[30,61]. One of the important questions is whether the sulfur reduction follows a stepwise mechanism[10], or is dominated by multiple parallel pathways among the sulfide/polysulfide species[44]. Our results provide evidence of the non-synchronous changes of the long-chain and intermediate polysulfides, revealing the stepwise evolution mechanism during the multi-step polysulfide reduction process.

The recharge processes of the cathode reactions involve the reverse processes of the discharge, where the $Li_2S$ is oxidized to polysulfides and subsequently to sulfur through solid-liquid-solid transition processes. The voltage profile is shown in Fig. 5a, while Fig. 5b presents the Raman spectra at different states of charge (SOCs). Peaks at 400 cm⁻¹ and 453 cm⁻¹ simultaneously increased, reached their maximum values at 42% SOC (around the start of the plateau at 2.4 V), and then decreased, showing the consistent reaction trends of long-chain and intermediate polysulfides. The on surface (2-D) mapping images of polysulfide evolution and the corresponding quantified changes are shown in Figs. 5c, d, followed by the in depth (Z-direction) results in Figs. 5e, f. Both long-chain and intermediate polysulfides show consistent trends throughout the entire recharge process, revealing a parallel evolution mechanism. These trends are clearly different from the stepwise discharge process, and can be mainly attributed to the sluggish activation process of $Li_2S$ when compared to the liquid-liquid transitions among polysulfides[42,62] during discharge. Figure 5e illustrates the continuous diffusion of the polysulfides. We speculate that the polysulfides diffuse into the non-aqueous liquid electrolyte solution during the entire discharge-recharge processes, providing direct evidence of the loss of active material and capacity

fade. Subsequently, sulfur clusters reformed on the electrode at around 95% SOC, with increasing size (as marked by circles) and new nuclei until the end of the charge (Fig. 5g). It should be noted that sulfur clusters, with larger sizes, may not be able to dissolve completely during the initial discharge process, because of their limited electronic conductivity. New ones would deposit onto the pristine sulfur clusters, causing the formation of "dead" (electronically disconnected) sulfur in subsequent cycles, as shown in Supplementary Fig. 25. The corresponding optical images are shown in Supplementary Fig. 26.

## Discussion
In summary, *operando* confocal Raman microscopy has been employed to systematically investigate the reaction pathways and kinetics of the multi-step Li−S redox processes, involving detailed analysis of reaction rates and polysulfide evolution during cell operation. Based on chronoamperometric measurements, the sub-steps of sulfur reduction and the polysulfide redox processes were both shown to follow first-order reaction kinetics, exhibiting clear dependence of their reaction rates on the electronic conductivity of sulfur and the concentration of the polysulfides. A direct correlation was also established between the morphologies of $Li_2S/S$ and their potential-dependent nucleation and growth processes. In addition, the real-time observation during galvanostatic reactions provided essential insights into the evolution mechanisms of different polysulfides and compelling evidence of their diffusion processes into the electrolyte. The analytical protocols and models presented here have enabled a clearer understanding of the mechanisms and kinetics of Li−S redox processes. These findings and approaches could, in turn, be used to identify potential performance improvements and extended to investigate consecutive reactions and complex pathways in other state-of-the-art electrical energy storage systems.

## Methods
### Li−S cells assembly
Lithium bis(trifluoromethanesulfonyl)imide (LiTFSI, anhydrous, 99.95%), 1,2-Dimethoxyethane (DME, anhydrous, 99.5%), 1,3-dioxolane (DOL, anhydrous, 99.8%), sulfur (99.5%) and polyvinylidene fluoride (Mn ~ 2000) were purchased from Sigma-Aldrich. $Li_2S$ and Li metal (0.75 mm, 99.9%) were from Alfa-Aesar. For Li−S coincells, sulfur electrodes were prepared by coating a slurry of sulfur: polyvinylidene fluoride (8:2) onto pinhole-modified carbon paper (AvCarb EP40, Fuelcell Store, 0.19 mm thickness). Furthermore, an additional Super P layer (Alfa-Aesar, 99%) was added to the dried sulfur layer, in an effort to investigate the effects of the active interfacial area on the kinetics of sulfur reduction. $Li_2S_4$ catholytes (60 μL), at various concentrations, were used to assemble Li−polysulfide coin cells by mixing stoichiometric amounts of $Li_2S$ and sulfur in 1.0 M LiTFSI in DOL/DME and allowing to stand at 60 °C overnight. The complex interconversion and interplay of the different polysulfides, as well as potential short-chain sulfide deposits, would affect the accuracy of our detection. Thus, the $Li_2S_4$ solution was used for the investigation of polysulfide reduction. Lithium metal was employed as simultaneous counter and reference electrode in the two-electrode coin cell system.

A 3 mm diameter hole was drilled on the cap of the coin cells casing (prior to cell assembly) for optical observation. A Raman transmissive film (glass, 0.13 to 0.17 mm, Fisherbrand) was then glued (two part epoxy, mix and wait for 5 min, Devcon) onto it to ensure a hermetic seal. The coin cells were assembled in an argon-filled glove box (Vacuum Atmospheres Co.) with oxygen and water levels below 0.1 ppm and 0.1 ppm, respectively.

### *Operando* confocal Raman microscopy characterization
The Raman spectra and mapping images were obtained with a WITec Alpha300R confocal Raman microscope (placed in a laboratory with an air environment and controlled temperature of 25 ± 1 °C). The Raman mapping was captured every 5 mins in a $150 \times 150$ μm$^2$ region. The calibration of the spectra and color contrast of the images were set consistently for quantification and comparison using the software of Project Five 5.1. 20 cm$^{-1}$ regions centered around the peak positions on the spectra were selected for mapping. Areas in the Raman mapping were extracted and quantified by ImageJ. For sulfur clusters, projected areas were mapped, which served as proxy for the active surface areas. For $Li_2S_4$ catholytes with different concentrations, the areas at open circuit were treated as pristine concentrations. Since the $Li_2S_4$ catholytes are homogeneous solutions, the changes of the areas from Raman mapping were assumed to be proportional to changes in concentration. In potentiostatic experiments, changes in areas were converted to the corresponding concentrations to plot vs time. *Operando* characterizations for potentiostatic and galvanostatic measurements were achieved by controlling the voltage/reaction rate of the cells and monitoring the current-time/voltage-time responses, respectively, with a potentiostat (Methrohm Autolab PGSTAT302N).

### Electrochemical experiments
Cyclic voltammetry (CV) was conducted on a Solartron electrochemical workstation at a sweep rate of 0.1 mV s$^{-1}$. Galvanostatic intermittent titration technique (GITT) measurements were performed using a Neware battery test station with a cutoff voltage of 1.5–3.0 V (vs. Li$^+$/Li). A constant current rate of 0.1 C (1.0 mA) was applied for 5 min, followed by a relaxation step of 1 h until the voltage reached upper or lower limits. Chronoamperometry experiments were performed using an Autolab potentiostat (Methrohm Autolab PGSTAT302N), to control the potentials of different cells and record the current-time transients. The tests were carried out at 25 ± 1 °C. No climatic/environmental chamber was used.

### Ex situ characterization
For investigation of the electrodeposition of $Li_2S$, the Li−polysulfide cells were assembled and the potentiostatic discharge processes performed at 2.0, 2.05 and 2.1 V. The positive electrode was carbon paper (AvCarb EP40, Fuelcell Store, 0.19 mm thickness). In the electrolyte of $Li_2S_4$ (0.5 M, 60 μL), at applied potentials of 2.0–2.1 V, the polysulfides in the electrolyte can be reduced to solid $Li_2S$ which deposits onto the carbon electrode, forming the positive electrode. The cells were disassembled in an Ar-filled glovebox with oxygen and water levels below 0.1 ppm and 0.1 ppm, respectively, after the corresponding potentiostatic tests. Cathode samples with the deposited $Li_2S$ on carbon papers were attached onto the sample holders (platforms that can load samples for observations of scanning electron microscopes, aluminum specimen mounts, slotted head, Electron Microscopy Sciences) and dried in vacuum (antechamber of the Ar-filled glovebox with oxygen and water levels below 0.1 ppm and 0.1 ppm, respectively, 25 ± 1 °C) for 2 h. They were then sealed in plastic poly bags filled with Ar and transported to the equipment (about 5 s exposure to air) for ex situ measurements. Morphologies and elemental maps of cathode samples were acquired with a field-emission scanning electron microscope (Zeiss Gemini 500, 0.75 eV).

### Reporting summary
Further information on research design is available in the Nature Research Reporting Summary linked to this article.

## Data availability
The authors declare that all experimental data and relevant analysis of this work are available from the corresponding author (HDA) upon reasonable request.

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

## Acknowledgements

The authors acknowledge support from the Cornell Energy System Institute (CESI), Daimler/Mercedes-Benz. This work made use of Raman and SEM facilities of the Cornell Center for Materials Research Shared Facilities which are supported through the NSF MRSEC program (DMR-1719875). S.L. thanks Philip Carubia for the help of Raman measurements.

## Author contributions

H.D.A. supervised the project. S.L. and H.D.A. conceived the research in the manuscript. S.L. performed the *operando* confocal Raman experiments and wrote the draft of the paper. S.-H.Y. and S.L. fabricated the cells with Raman transmissive windows. X.F. and S.L. designed the experimental setups. M.R.K., X.F., S-H.Y. and S.L. discussed and revised the manuscript under the guidance of H.D.A. All authors have confirmed the final version of the manuscript.

## Competing interests

The authors declare no competing interests.
