## [Peer Review File · Nature Communications]

REVIEWER COMMENTS

Reviewer #1 (Remarks to the Author):

This revised manuscript answers some of the criticisms from the first round. The authors now provide a description of the fairly copious prior work examining sulfur reduction kinetics in Li-S batteries, including some of the work using Raman spectroscopy. The authors find, in agreement with this prior work, that first order (or pseudo first order) kinetics apply and that there are suggestions of a step-wise reduction process. These step-wise processes have been suggested earlier by electrochemical and other means.

In this paper there is further development of the S₈ to Li₂S reaction mechanism. A transition between film to island is found in the Li₂S deposit, in agreement with prior electron and other microscopy work. Also in agreement with copious prior work, Li₂S nucleation and growth mechanisms are imaged and discussed. Also in agreement with prior efforts, including but not limited to those involving Raman microscopy, the authors show that different length polysulfides exhibit different behaviors in the electrolyte.

This paper brings together much of this prior effort in one place and shows the power of Raman microscopy to address Li-S battery cathodes.

Reviewer #2 (Remarks to the Author):

The study is certainly interesting and the data and insights gained from it are useful for the advancement to the Li-S field. The authors did a good job of addressing many of my original comments. Although many operando Raman experiments have been done on the Li-S system, the imaging component of the current manuscript is interesting. The dependence of Li₂S morphology on the overpotential applied is a new insight. The kinetic treatment is thorough and a useful addition to the field. It's becoming more and more rare for a Li-S paper to provide any fundamental insight into the mechanisms and this paper will be one of the few.

I still suggest, however, that the data does not support the conclusion that the Li-S discharge mechanism is stepwise. A stepwise mechanism would mean that first long chain polysulfides form followed by a decrease in long chain polysulfides with a concurrent increase in short chain (or intermediate), and these changes should track with the plateaus in the electrochemistry. That is not what is observed here. Long chain and intermediate chain polysulfides increase together and decrease together, although they peak at different points in the curve. I sincerely believe that describing the mechanism as a stepwise mechanism is an over-simplification of the mechanism.

Reviewer #3 (Remarks to the Author):

In the manuscript entitled “Mechanism and Kinetics of Lithium-Sulfur Redox via Operando Confocal Raman Microscopy», the authors report on an investigation into the mechanism of Li-S battery conversion reactions employing Raman Microscopy.

The paper is very extensive and provides investigations in a systematic manner, first dealing with the potentiostatic reduction of sulfur to polysulfides, then with their further reduction, precipitation of Li_2S and oxidation of polysulfides to sulfur. The second part then follows the reaction under galvanostatic conditions during both discharge and charge.

I have several specific questions and comments regarding the presented work:

1. In the beginning of the Results and discussion section, the authors provide the main sub-steps by giving a single reduction reaction originating from elemental sulfur and producing polysulfides of various chain length. I am not sure what this part of the discussion is supposed to signify? Do author claim that all polysulfides are formed in a reaction from elemental sulfur, i.e., no polysulfides are formed by reduction of more oxidized polysulfide species? Because this assumption is contradictory to the state-of-the-art knowledge. At this point in the manuscript, such a claim is also not substantiated by any experiments. Even if I have understood this wrong and the authors meant this part of the manuscript to serve as a general basis for further discussion, then there should be more commentary included with it. I find it wrong to try to minimize the extent of the complexity of the redox processes in between sulfur and polysulfide species in a paper, which is specifically supposed to “elucidate the underlying mechanism of polysulfide generation/evolution». There are a lot more reactions taking place both through direct reduction of different polysulfides and through disproportionation and comproportionation reactions.
2. The authors should pay more attention to their phrasing of long- or intermediate- or short-chain polysulfides. The following statement: »In the initial reduction process, two cathodic peaks at 2.30 and 1.97 V are observed, corresponding, respectively, to the reduction of sulfur to soluble long-chain polysulfides and intermediate products to Li_2S .« reads as if in the first peak only polysulfides with a

chain length between 6 and 8 sulfur atoms are formed. This is wrong, as numerous HPLC studies on the detection of polysulfides have proven that shorter-order polysulfides are also formed in the initial sulfur reduction reaction.

3. The Raman peak analysis for different polysulfide chain lengths is confusing. The authors first state that the 400 cm^{-1} band belongs to the central S-S bonds and the 450 cm^{-1} to the terminal S-S bonds. To me that means that all polysulfides with central S-S bonds will have a band in the vicinity of 400 cm^{-1} , while the authors limit this to polysulfide with the chain length between 6 and 8. Why? Similarly, a band for terminal S-S bond should be present in all polysulfide species except for Li_2S ?

4. I have checked the paper cited for the attribution of bands to specific modes and that paper calculated the most intense band for S_6^{2-} and S_8^{2-} polysulfides to be at 320 cm^{-1} . Why is this band not seen in your data?

5. When adding a carbon interlayer the reaction rate increased 7-times, but you mostly limited this information to the supporting info. I find it important enough to be included in the main paper. This is not negligible.

6. The analysis of the nucleation and growth mechanism by fitting of 3- or 2-dimensional theoretic peaks is completely wrong for several reasons:

a. The equations employed have been developed for flat electrodes, they do not work on non-planar electrodes such as employed in your study. Yes, other papers on Li-S have reported using the same equations – if they used porous carbon electrodes or carbon papers for the electrode basis, the equations simply do not hold.

b. When starting with polysulfide catholyte and applying a potentiostatic step reduction, the current response is not only due to nucleation and growth (and double-layer charging), but due to background polysulfide reduction to shorter chain polysulfides. This background should not be disregarded and has to be subtracted before attempting any peak fitting.

c. Specific proof of discrepancies between the data – for the reduction of polysulfides at 2.05 V vs. Li/Li^+ , you determined 2D deposition. This means that only a very thin (a quasi monolayer) of Li_2S is supposed to form on the electrode surface and that its further formation is limited due to the nonconductive deposit passivating the surface. Such a deposit can only be a few nm thick by default. Yet the SEM data shows very thick (on the order of μm) deposits on the carbon fibers.

7. Your comment on whether the mechanism of operation is through a stepwise mechanism or through parallel pathway is not convincing. Partly because the attribution of bands to different polysulfide species is questionable (points 3. and 4.) and partly because it is barely discussed. Does this mean that the authors doubt the presence of disproportionation reactions? The conclusion on the stepwise versus parallel pathway could not be made with the data presented accounting to the short-comings of the techniques/methods employed.

8. I would wish for more discussion regarding the in-depth mapping of the cross-sectional distribution of polysulfides. On the image it seems that the polysulfide signal disappears even before reaching the anode. Is this a consequence of dilution? Can this be used to determine disproportionation reactions between polysulfides?

Reviewer #4 (Remarks to the Author):

The work presented in this manuscript explores the use of operando confocal Raman microscopy to gain mechanistic insight into polysulfide formation kinetics. As evident from this piece of work and previous reports exploring the use of Raman spectroscopy or other operando/in-situ techniques to understand the mechanism and kinetics of lithium-sulphur redox, it is clear that unambiguous elucidation of the discrete mechanistic steps is extremely challenging. The reduction reactions of sulphur, including its transformation to long-chain polysulfides and intermediate-chain polysulfides, have been discussed in many theoretical and experimental articles already. The novelty of this work is indeed the use of Raman microscopy in this context, and this work further supports the existing idea of polysulfide formation and reaction kinetics (arguably first order as suggested in some previous reports as well). In fact, the use of other microscopy techniques (in-operando transmission X-ray microscopy, in-situ X-ray fluorescence microscopy, electron microscopy, etc) in this context have also been reported. While this study nicely demonstrates the use of Raman microscopy and provides additional evidence to the pool of information available on this topic in the literature, I believe there's no notable new insight or an unambiguous elucidation of the reaction mechanism based on this new approach that would make it suitable for Nature Communications. The use of operando confocal microscopy in this context is interesting, but this manuscript is more suitable for a more specialized journal in my opinion.

Point-by-point response

Below we include the reviewers' comments in Black, and our responses (in Blue).

Reviewer #1 (Remarks to the Author):

This revised manuscript answers some of the criticisms from the first round. The authors now provide a description of the fairly copious prior work examining sulfur reduction kinetics in Li-S batteries, including some of the work using Raman spectroscopy. The authors find, in agreement with this prior work, that first order (or pseudo first order) kinetics apply and that there are suggestions of a step-wise reduction process. These step-wise processes have been suggested earlier by electrochemical and other means.

We thank the reviewer for the comments and for recognizing and appreciating the findings discussed in our work. Examining the complex sulfur reduction kinetics and reaction mechanism have long been regarded as some of the most important but challenging issues that govern the further progress of Li-S batteries. The difficulties in the mechanistic understanding of the Li-S system lie not only in the multistep reduction process of sulfur itself, involving solid-liquid-solid transformations and many intermediate species, but also the specific identification, possible visualization, and quantification of the various (poly)sulfide species with different lengths, in real-time, during cell operation. In the field of Li-S research, there is a significant body of work employing various advanced characterization techniques to distinguish sulfur/polysulfides/Li₂S, and compare the reaction pathways among various modified systems, like host optimized/interlayer added cathode materials, solvent/salt/additive mediated electrolytes, and SEI improved anodes (Quan-Hong Yang et al. *Energy Environ. Sci.*, **2017**, *10*, 1694; Jang Wook Choi et al. *Adv. Energy Mater.*, **2020**, 2001456; Liqiang Mai et al. *Nanoscale*, **2017**, *9*, 19001, etc.). There is a much smaller number of publications focused on the fundamental insights into the Li-S redox mechanisms, such as the reaction order, potential-dependent reaction rates, and transformation mechanism and kinetics among the different polysulfides. Although a number of studies have been devoted to these efforts, unambiguous and thorough treatment and elucidation of the sulfur reduction kinetics employing appropriate *operando* characterization tools remain incomplete. We believe that the work presented in our manuscript is a significant advance of our understanding of Li-S redox kinetics employing confocal Raman imaging. We have carefully read and analyzed the prior Raman work that the reviewer mentioned (Gewirth et al. *ACS Appl. Mater. Interfaces*, **2015**, *7*, 1709). The novelty and differences of our work compared with the previous work are highlighted in detail below.

1. Developed Raman imaging techniques using *operando* confocal Raman microscopy. Confocal Raman microscopy combines the strengths of both Raman spectroscopy that enables simultaneous identification of sulfur and various polysulfides, and confocal microscopy that enables the high-resolution imaging of the exact focal plane, which are ideally suited to investigate the important issues of Li-S mechanisms involving the multi-step redox processes and solid-liquid-solid transformations. In our work, we visualized sulfur and polysulfides of different lengths (short, medium and long) and further monitored their spatial distribution and evolution during potentiostatic and galvanostatic redox processes. The transport of soluble polysulfides away from the cathode via diffusion is explicitly captured by depth mapping from the cathode surface into the electrolyte. The outstanding temporal performance of *operando* Raman imaging techniques (capturing a 150×150 μm² every 5 mins), specific and simultaneous identification of different sulfide species, and spatial resolution down to submicron level, open a new window for us to carefully observe the Li-S redox processes and lay a solid foundation for in-depth analysis.

2. Systematically analyzed the kinetics of sulfur reduction and polysulfide redox processes. On the basis of precise visualization and quantification of the sulfur clusters and polysulfides, and comparison among various models, we demonstrated the first-order kinetics and calculated the potential-dependent reaction rates of sulfur reduction and polysulfide redox. The effects of the electronic conductivity of the sulfur clusters and the concentrations of the polysulfides have been characterized, reflecting the important

correlation of the interfacial phenomena and cell performance. In addition, we gained valuable new insights into the dependence of the nucleation and growth mechanisms of sulfur and Li_2S on the applied overpotentials, which demonstrated clear differences in their electrodeposition processes and could help inform novel approaches to modify the liquid-solid transformations in advanced Li-S systems. In prior work, the authors directly applied a first-order kinetics analysis to fit the sulfur reduction data. The simple combination of the multi-step redox processes using high overpotentials and the lack of analysis focusing on polysulfides lead them to conclude that short-chain S_3^{2-} is formed directly from sulfur decomposition, which does not agree with the generally accepted views about the solid-liquid-solid transformations and the initial formation of long-chain polysulfides during sulfur reduction. The step-by-step experiments and systematic analysis, based on Raman imaging in our work, provide compelling evidence and have provided a more thorough understanding of the sulfur reduction kinetics in Li-S batteries.

3. Identified different polysulfides and investigated their reaction mechanisms during discharge and charge processes. In the prior work, the authors observed numerous (poly)sulfide species during Li-S redox using *in situ* Raman spectroscopy. No further discussion was provided in terms of the trends of their evolution, which can be attributed, at least in part, to the limited accuracy of the spectroscopic peaks for quantification, and the very localized measurement of only one spot on the cathode surface. The transformation mechanisms between/among the different polysulfides remain elusive. In our work, different polysulfides were identified and individually visualized during the discharge and charge processes. The Raman images captured on the cathode surface, and deep into the electrolyte, clearly revealed their spatial distribution and differences in intensity changes at different states of discharge/charge, suggesting a stepwise reduction process but parallel oxidation mechanism during cell operation. These observations provide important clues and insights for the discussion of the solid-liquid-solid transformation mechanism and kinetics. Our observations enhanced the understanding of polysulfide redox and can inform the design of specific and high activity catalysts in the Li-S field.

Therefore, we strongly believe that our present work is not a simple description of prior suggestions, but rather a successful implementation of confocal Raman methods that have provided compelling experimental evidence of sulfur reduction kinetics and mechanisms that extend our knowledge of Li-S redox processes. Moreover, they stimulate further exploration of Raman imaging to investigate multiple complex reaction processes. These comments and related explanations have been incorporated into the revised manuscript (Pages 3-4).

In this paper there is further development of the S_8 to Li_2S reaction mechanism. A transition between film to island is found in the Li_2S deposit, in agreement with prior electron and other microscopy work. Also in agreement with copious prior work, Li_2S nucleation and growth mechanisms are imaged and discussed.

We thank the reviewer for the comments and observations. While many prior works imaged different deposited morphologies of Li_2S and discussed the Li_2S nucleation and growth mechanisms, the effect of the most fundamental, decisive factor of Li_2S electrodeposition, mainly the driving force, has not been clearly articulated nor understood up to date. In prior works, the authors employed the deposited morphologies and the nucleation mechanisms of Li_2S to compare the performance of various modified Li-S systems. The transformations from film to island morphology, and/or 2D to 3D nucleation and growth models were attributed to various external factors such as the host materials, the catalysts, and the solvent and/or salt. These factors can directly affect the equilibriums of Li_2S electrodeposition reactions and the exact overpotentials applied, which, undoubtedly, should be taken into consideration in the discussions. The careful and rigorous investigation of the correlations between/among the overpotentials, nucleation and growth models, deposited Li_2S morphologies, and the corresponding capacities presented in our work, have directly addressed these issues and have revealed new insights into the Li_2S electrodeposition mechanisms, which is crucial for the fundamental understanding and improved design

of next-generation Li-S batteries. We have expanded this part of the narrative in the revised manuscript (Page 15).

Also in agreement with prior efforts, including but not limited to those involving Raman microscopy, the authors show that different length polysulfides exhibit different behaviors in the electrolyte.

We thank the reviewer for the comments and for emphasizing our study of polysulfide behaviors during Li-S redox. Note that a quantitative, accurate characterization and comprehensive investigation of the redox kinetics and pathways of different polysulfides requires the joint efforts of the simultaneous identification and high-resolution imaging of them in real-time during discharge and charge processes, which is difficult to achieve and has not been systematically reported by prior works including but not limited to the ones involving Raman microscopy. The systematic and quantitative insights and understanding presented in our work, including the redox kinetics, the potential- and concentration-dependent reaction rates, the transformation mechanisms, and the diffusion processes of different polysulfides, are unique and important accomplishments, adding fresh impetus to the improvement of Li-S research. These comments have been added to the revised version of the manuscript (Pages 3-4 and 23).

This paper brings together much of this prior effort in one place and shows the power of Raman microscopy to address Li-S battery cathodes.

We appreciate the reviewer's comments and for acknowledging the novelty of our work in showing the power of Raman microscopy to address the complex Li-S redox processes. As we have articulated above, we sincerely feel that the work presented in our manuscript provides compelling evidence, essential insights, and useful methods for Li-S investigation, addressing unresolved issues of fundamental importance to Li-S battery research. These, in turn, can inform and inspire enhanced, high-performance design of next-generation Li-S batteries. We have extended our narrative and discussion in the revised manuscript (Pages 3-4, 15 and 23).

Reviewer #2 (Remarks to the Author):

The study is certainly interesting and the data and insights gained from it are useful for the advancement to the Li-S field. The authors did a good job of addressing many of my original comments. Although many operando Raman experiments have been done on the Li-S system, the imaging component of the current manuscript is interesting. The dependence of Li₂S morphology on the overpotential applied is a new insight. The kinetic treatment is thorough and a useful addition to the field. It's becoming more and more rare for a Li-S paper to provide any fundamental insight into the mechanisms and this paper will be one of the few.

I still suggest, however, that the data does not support the conclusion that the Li-S discharge mechanism is stepwise. A stepwise mechanism would mean that first long chain polysulfides form followed by a decrease in long chain polysulfides with a concurrent increase in short chain (or intermediate), and these changes should track with the plateaus in the electrochemistry. That is not what is observed here. Long chain and intermediate chain polysulfides increase together and decrease together, although they peak at different points in the curve. I sincerely believe that describing the mechanism as a stepwise mechanism is an over-simplification of the mechanism.

We thank the reviewer for the comments and the recognition of our work. For the polysulfide stepwise mechanism, we have further explained our use of the term "stepwise" in the revised manuscript. Briefly, despite the fact that the concentration of long and intermediate chain polysulfides simultaneously increased and decreased in the earlier and latter stages of discharge, the decrease of the long chain and

the increase of the intermediate chain polysulfides in the middle region can be regarded as a stepwise discharge process. Our considerations are mainly based on the two arguments discussed below. 1. The polysulfides with different lengths show different changes in intensity trends. In the regions between the peaks, the decrease of the long-chain and the increase of the short-chain polysulfides indicate the transformation from long-chain to short-chain, which is key to the evolution of short-chain polysulfides during discharge. The prominent function of the intermediate products reveals the characteristics of the stepwise reaction. 2. The stepwise discharge process distinguishes itself from the parallel recharge process in which both long-chain and short-chain polysulfides show consistent trends during the whole recharge process.

Moreover, it is worth mentioning that the longer-than-expected increase region of the long-chain polysulfides can be due, at least in part, to their slower diffusion processes relative to the short-chain ones and the sluggish solid-liquid reduction from sulfur, when compared to the liquid-liquid transformations between polysulfides. More detailed explanations and revisions have been added to the revised manuscript (Pages 22-23, 26) and SI (Page 25).

Reviewer #3 (Remarks to the Author):

In the manuscript entitled “Mechanism and Kinetics of Lithium-Sulfur Redox via Operando Confocal Raman Microscopy», the authors report on an investigation into the mechanism of Li-S battery conversion reactions employing Raman Microscopy.

The paper is very extensive and provides investigations in a systematic manner, first dealing with the potentiostatic reduction of sulfur to polysulfides, then with their further reduction, precipitation of Li₂S and oxidation of polysulfides to sulfur. The second part then follows the reaction under galvanostatic conditions during both discharge and charge.

I have several specific questions and comments regarding the presented work:

1. In the beginning of the Results and discussion section, the authors provide the main sub-steps by giving a single reduction reaction originating from elemental sulfur and producing polysulfides of various chain length. I am not sure what this part of the discussion is supposed to signify? Do author claim that all polysulfides are formed in a reaction from elemental sulfur, i.e., no polysulfides are formed by reduction of more oxidized polysulfide species? Because this assumption is contradictory to the state-of-the-art knowledge. At this point in the manuscript, such a claim is also not substantiated by any experiments. Even if I have understood this wrong and the authors meant this part of the manuscript to serve as a general basis for further discussion, then there should be more commentary included with it. I find it wrong to try to minimize the extent of the complexity of the redox processes in between sulfur and polysulfide species in a paper, which is specifically supposed to “elucidate the underlying mechanism of polysulfide generation/evolution». There are a lot more reactions taking place both through direct reduction of different polysulfides and through disproportionation and comproportionation reactions.

We thank the reviewer for the insightful comments and observations. At the beginning of the Results section, using the reduction equations, we intended to show the main sulfur reaction pathways and provide a general basis for further step-by-step discussions. We do believe that polysulfides can be formed by the reduction of more oxidized polysulfide species. We have indeed observed the stepwise transformation from long-chain to intermediate polysulfides from the galvanostatic discharge experiments (Figure 5), which show clear and compelling evidence in support of our assertions. In an effort to be clearer, we have modified our expression to reveal the ring-opening and chain-shortening processes during sulfide reactions and included more commentary with it. We have revised the expression as follows,

More specific reaction equations, including some typical reduction sub-steps and possible disproportionation and conproportionation reactions, are presented in the revised Supplementary Table 2.

Supplementary Table 2. Main sub-steps of the sulfur reduction process and accompanying disproportionation and conproportionation reactions.

Stages	Proposed reactions
Solid-liquid reduction from sulfur to polysulfides	$S_8 + 2e^- \rightarrow S_8^{2-}$
Polysulfide transformation	$S_8^{2-} + 2e^- \rightarrow 2S_4^{2-}$ $S_6^{2-} + e^- \rightarrow 3/2S_4^{2-}$
Formation of short-chain sulfides	$S_4^{2-} + 2e^- \rightarrow 2S_2^{2-}$ $S_2^{2-} + 2e^- \rightarrow 2S^{2-}$
Disproportionation and conproportionation reactions	$S_8^{2-} \rightarrow 1/4S_8 + S_6^{2-}$ $S_6^{2-} \rightarrow 2S_3^{2-}$ $S_4^{2-} \rightarrow 3/5S_6^{2-} + 2/5S_2^{2-}$ $3/5S_6^{2-} + 1/5S_2^{2-} \rightarrow S_4^{2-}$

In addition, we have explained, more in detail, the disproportionation and conproportionation reactions. The disproportionation and conproportionation reactions of the polysulfides can easily take place and keep dynamic equilibriums in solution during the entire discharge and charge processes. In an effort to confirm the accuracy of our detection and exclude side reactions involving the disproportionation and conproportionation reactions from our observations, we have investigated the Raman signals of the polysulfides at open circuit. As shown in Supplementary Figure 10, the signals remained stable for extended time periods, indicating that the equilibria of the dis- and con- proportionation reactions are established rapidly and that our detection would not be affected by the complex side reactions. Therefore, we focused on the discussion of the reduction reactions in our manuscript. These comments and explanations have been added to the revised manuscript (Pages 5 and 10-11) and SI (Page 3).

2. The authors should pay more attention to their phrasing of long- or intermediate- or short-chain polysulfides. The following statement: »In the initial reduction process, two cathodic peaks at 2.30 and 1.97 V are observed, corresponding, respectively, to the reduction of sulfur to soluble long-chain polysulfides and intermediate products to Li_2S .« reads as if in the first peak only polysulfides with a chain length between 6 and 8 sulfur atoms are formed. This is wrong, as numerous HPLC studies on the detection of polysulfides have proven that shorter-order polysulfides are also formed in the initial sulfur reduction reaction.

We appreciate the reviewer for the comments. We agree with the reviewer that the polysulfides with different lengths co-exist in the solution. In the revised manuscript, we have added more citations about this point, including some HPLC studies (Celine Barchasz et al. *Anal. Chem.*, **2012**, *84*, 3973–3980; Deyang Qu et al. *J. Power Sources*, **2016**, *301*, 312–316), and modified our statement as follows to make our arguments clearer. “In the initial reduction process, two cathodic peaks at 2.30 and 1.97 V are observed, corresponding, respectively, to the reduction of sulfur to soluble polysulfides and their further reduction forming Li_2S .” (Page 5)

3. The Raman peak analysis for different polysulfide chain lengths is confusing. The authors first state that the 400 cm^{-1} band belongs to the central S-S bonds and the 450 cm^{-1} to the terminal S-S bonds. To me that means that all polysulfides with central S-S bonds will have a band in the vicinity of 400 cm^{-1} , while the authors limit this to polysulfide with the chain length between 6 and 8. Why? Similarly, a band for terminal S-S bond should be present in all polysulfide species except for Li_2S ?

We thank the reviewer for the comments and apologize for the confusion. Raman spectroscopy has been essential for the identification of polysulfide species since the 1960s. In 1988, P. Dubois et al. reported the first Raman spectra of solutions of lithium polysulfides in liquid ammonia. Raman spectra of Li_2S_4 solutions showed a very strong peak around 430 cm^{-1} . As the chain lengths increased

from 4 to 5 and 6, this peak decreased and was accompanied by the formation and gradual increase of a peak at 400 cm^{-1} . In a typical Raman spectrum of a Li_2S_6 solution, the peak at 400 cm^{-1} is the most intense one (J. P. Lelieur et al. *Inorganic Chemistry*, **1988**, 27, 73-80).

In more recent times, M. Hagen et al. investigated different polysulfides by a combination of calculations and *in situ* experiments in Li-S cells. Results from the calculations showed that for a polysulfide di-anion with a specific chain length of n (S_n^{2-}), the Raman peaks in the 350-500 cm^{-1} region could be identified as S-S stretch vibrations, involving the central S-S bonds and terminal S-S bonds with the terminal modes always shifted to higher wavenumbers. Note that it is a distribution of polysulfides with different chain lengths, rather than one kind of polysulfide that exists in polysulfide solutions. In addition, the Raman positions of the peaks can vary by several cm^{-1} with different polysulfide compositions and different solvents. During the *in situ* Raman measurements of the Li-S cell, the authors detected two broad peaks over the ranges of 340-420 cm^{-1} and 420-480 cm^{-1} in the middle stage of the discharge process and attributed them to mixtures of longer polysulfides of S_6^{2-} , S_7^{2-} , S_8^{2-} and shorter polysulfides of S_3^{2-} , S_4^{2-} , S_5^{2-} , respectively, according to the nature of sulfur reduction process and results from calculations. (M. Hagen et al. *Journal of The Electrochemical Society*, **2013**, 160, A1205-A1214). This identification is in good agreement with many other experimental reports (Qiang Zhang et al. *Small Methods*, **2017**, 1, 1700134;

Supplementary Figure 10. Operando confocal Raman investigation of Li – polysulfide cells at open circuit. (a) Plot of the potential changes at open circuit. *Operando* Raman (b) spectra and (c-e) mapping images during cell storage. The blue color in (c-e) shows the signal from intermediate polysulfides, centered at 453 cm^{-1} with a band width of 20 cm^{-1} . The color contrast is consistent in images and shown under (e). (f) Raw data of the area changes of the mapping results during cell storage.

K. Zaghib et al. *Sustainable Energy Fuels*, **2017**, *1*, 737), widely employed in Li-S investigations that use Raman characterization.

In our experiments, we observed two peaks at 400 and 450 cm^{-1} in polysulfide solutions, and employed them to represent long-chain Li_2S_x ($x=6-8$) and intermediate-chain Li_2S_x ($x=3-5$) species in polysulfide solutions, respectively. Our observations revealing the evolution of the polysulfides can, in turn, help identify and inform our mechanistic attributions, mainly, the stepwise changes of these two peaks during the galvanostatic discharge process (Figure 5), and the faster decrease of the 400 cm^{-1} peak during the chronoamperometric experiment at 2.0 V (Figure 2 and Supplementary Figure 13). In an effort to provide a clearer statement, we have changed our description as follows. "According to previous calculations and experimental reports, we employed these two peaks to represent long-chain Li_2S_x ($x=6-8$) and intermediate-chain Li_2S_x ($x=3-5$) polysulfides, respectively." These modifications and corresponding citations have been added to the revised manuscript (Page 7).

4. I have checked the paper cited for the attribution of bands to specific modes and that paper calculated the most intense band for S62- and S82- polysulfides to be at 320 cm^{-1} . Why is this band not seen in your data?

We thank the reviewer for the comments and observations. We have carefully read and checked the computational details of the cited paper (Patrik Johansson et al. *ChemPhysChem*, **2015**, *16*, 2755–2759). These calculations employed diethyl ether as the solvent and various dielectric constants to mimic the practical electrolyte solutions. As we discussed above, the positions of the Raman peaks will vary with the nature of the solvent(s) employed. In addition, the simplification/approximation and optimization procedures employed during the calculations can give rise to further deviations between the calculated results and the experimental data. In this cited report, and employing similar solvents, the authors also did not observe peaks at 320 cm^{-1} in their *operando* Raman measurements. They related the peak at 369 cm^{-1} , which can be detected in experiments, to the one at 320 cm^{-1} from the calculations. In our manuscript, we employed the typical solvents of DOL/DME. The Raman spectra that we acquired were in very good agreement with previous experimental reports (M. Hagen et al. *Journal of The Electrochemical Society*, **2013**, *160*, A1205-A1214; Qiang Zhang et al. *Small Methods*, **2017**, *1*, 1700134; K. Zaghib et al. *Sustainable Energy Fuels*, **2017**, *1*, 737). While we feel that calculations are very powerful tools to simulate and guide experiments for identifying general rules and elucidating mechanisms, it is difficult and impractical (and at times even risky) to attempt to precisely match the calculation results with the experimental data. We have changed our narrative and cited more experimental reports to more clearly explain this point in the revised version of the manuscript (Page 7).

5. When adding a carbon interlayer the reaction rate increased 7-times, but you mostly limited this information to the supporting info. I find it important enough to be included in the main paper. This is not negligible.

We thank the reviewer for the comments and for recognizing the importance of our discoveries. We have incorporated our results of the cell with the interlayer into the revised Figure 1 and put more emphasis on the discussion of the conductivity-dependent reaction rates of sulfur reduction.

Figure 1j provides the calculated rate constants at 2.30 V ($1.02 \times 10^{-4} \pm 0.02 \times 10^{-4} \text{ s}^{-1}$), 2.20 V ($2.16 \times 10^{-4} \pm 0.05 \times 10^{-4} \text{ s}^{-1}$), and 2.20 V using the cathode with an additional carbon interlayer ($7.17 \times 10^{-4} \pm 0.33 \times 10^{-4} \text{ s}^{-1}$). It shows that the value of the rate constant with the carbon interlayer increased by 3.5 times relative to the one without it, indicating that the enhancement of the electronic conductivity plays a significant role in increasing the density of active sites and accelerating the reaction rates of sulfur redox. This emphasizes the importance of micro-/nano- encapsulation of sulfur with a conductive matrix for achieving high percent utilization, which, in turn, could enhance overall device performance. These comments have been added to the revised manuscript (Pages 6-7 and 9-10).

Figure 1. Reduction of sulfur clusters at 2.30 V vs. Li⁺/Li. (a) Schematic illustration of the *operando* confocal Raman microscopy experimental setup for probing the mechanism and kinetics of Li-S redox processes. (b) Chronoamperometric current-time transient of sulfur reduction at 2.30 V (vs. Li⁺/Li). *Operando* Raman (c) spectra and (d-h) mapping images of the sulfur electrode during reduction. The red and yellow colors in (d-h) represent sulfur and long-chain polysulfides, respectively, where the color contrasts remained consistent for quantification. (i) Plot of the area changes of sulfur clusters with time. Inset: Linear fitting of the logarithm of the area with time according to first-order kinetics, $R^2=0.979$. (j) Values of k_s acquired at 2.30 V, 2.20 V, and 2.20 V using the cathode with an additional carbon interlayer.

6. The analysis of the nucleation and growth mechanism by fitting of 3- or 2-dimensional theoretic peaks is completely wrong for several reasons:

a. The equations employed have been developed for flat electrodes, they do not work on non-planar electrodes such as employed in your study. Yes, other papers on Li-S have reported using the same

equations – if they used porous carbon electrodes or carbon papers for the electrode basis, the equations simply do not hold.

We respectfully disagree with the reviewer on this point. We have carefully investigated the classical theories analyzing the kinetics of nucleation and growth. We found that the geometry of the substrate, whether it is planar or not, should not be a limitation to the application of the theory. In the early days, scientists employed planar substrates because they are easier to define. Performing typical 2D/3D electrodeposition processes on them served as simple experimental supports for computational simulation and complex equation derivations. The most important and fundamental equation to analyze the nucleation and growth process is the Avrami equation (Melvin Avrami, *J. Chem. Phys.*, **1939**, *7*, 1103). The core key for interpreting the Avrami equation lies in the expression below:

$$\frac{dV}{dV_{1ex}} = 1-V \quad (2)$$

where V is the actual extended volume of the new phase, V_{1ex} is the total extended volume without overlap, and 1-V is the ratio of untransformed matter. This equation means that the fraction of the extended volume that forms during each time increment that is real will be proportional to the volume fraction of untransformed matter. The BFT and SH models that we applied in our manuscript can be regarded as extensions of the Avrami equation in the field of electrochemical nucleation and growth processes, where the aim is to explain the i-t curves under different potentials (driving force), instead of the V-t curves under different temperatures as in the Avrami equations. Similarly, during electrodeposition processes, the increase in the actual current will be proportional to the fraction of the area uncovered. On the basis of this knowledge, two classical theories were then established, known as the BFT model for 2D and SH model for 3D nucleation and growth processes (A. Bewick et al. *Transactions of the Faraday Society*, **1962**, *58*, 2200-2216; Benjamin Scharifker et al. *Electrochimica Acta*, **1983**, *28*, 879-889). The corresponding expressions, with the dimensional and dimensionless forms, are provided in the revised Supplementary Table 3. We do believe the structure of the substrate will influence the values of many parameters in the equations, including the number density of active sites and rate constants of nucleation and growth. However, the shape of the curve indicating the specific nucleation process will not be affected by these quantities, as shown in the dimensionless expressions presented in Supplementary Table 3.

The dimensionless expressions reveal that the detailed interpretation of the curve is determined by the time dependence of the nucleation and growth rates, and by the geometry of the growing phase. Therefore, we do not believe that these theories should be limited to planar electrodes. In fact, most nucleation processes are not limited to 2D dimensions, such as cobalt electrodeposition on different substrates, zinc electrodeposition on steel, and Li metal nucleation and growth processes (M. Palomar-Pardave' et al. *Electrochimica Acta*, **2005**, *50*, 4736-4745, *Journal of Electroanalytical Chemistry*, **2002**, *521*, 95-106; G. Trejo et al. *Electrochimica Acta*, **2007**, *52*, 3686-3696; Bing-Joe Hwang et al. *J. Am. Chem. Soc.*, **2019**, *141*, 18612-18623, etc.). The wide application of the classical theories greatly facilitates the in-depth investigations of various nucleation and growth mechanisms. We have provided these explanations and equations in the revised manuscript (Pages 15-16) and SI (Pages 20-21).

Supplementary Table 3 Expressions of BFT and SH models.

Models	Expressions
BFT instantaneous nucleation	$I = \frac{2zF\pi MhN_0k^2t}{\rho} \exp\left(-\frac{\pi N_0M^2k^2t^2}{\rho^2}\right)$ $\frac{I}{I_{max}} = \frac{t}{t_{max}} \exp\left[-\frac{1}{2}\left(\frac{t^2}{t_{max}^2} - 1\right)\right]$

BFT progressive nucleation

$$I = \frac{zF\pi M h k^2 A N_0 t^2}{\rho} \exp\left(-\frac{\pi M^2 k^2 A N_0 t^3}{3\rho^2}\right)$$
$$\frac{I}{I_{max}} = \frac{t^2}{t_{max}^2} \exp\left[-\frac{2}{3}\left(\frac{t^3}{t_{max}^3} - 1\right)\right]$$

SH instantaneous nucleation

$$I = \frac{zFD^{1/2}c}{\pi^{1/2}t^{1/2}} [1 - \exp(-N\pi kDt)]$$
$$\frac{I^2}{I_{max}^2} = 1.9542 \frac{t_{max}}{t} \left[1 - \exp\left(-1.2564 \frac{t}{t_{max}}\right)\right]^2$$

SH progressive nucleation

$$I = \frac{zFD^{1/2}c}{\pi^{1/2}t^{1/2}} [1 - \exp(-AN_{\infty}\pi k'Dt^2/2)]$$
$$\frac{I^2}{I_{max}^2} = 1.2254 \frac{t_{max}}{t} \left[1 - \exp\left(-2.3367 \frac{t^2}{t_{max}^2}\right)\right]^2$$

where zF ($C \text{ mol}^{-1}$) is the molar charge, $F=96485 \text{ C mol}^{-1}$ is Faraday's constant, k is the growth rate constant ($\text{mol cm}^{-2} \text{ s}^{-1}$), h (cm) is the layer thickness, N_0 (cm^{-2}) is the number density of isolated centers, M (g mol^{-1}) is the molecular weight, and ρ (g cm^{-3}) is the density of the deposited material, A (s^{-1}) is the nucleation rate constant, N_{∞} is the number density of active sites, c (mol cm^{-3}) is the molar concentration, and D_0 ($\text{cm}^2 \text{ s}^{-1}$) is the effective diffusion coefficient.

b. When starting with polysulfide catholyte and applying a potentiostatic step reduction, the current response is not only due to nucleation and growth (and double-layer charging), but due to background polysulfide reduction to shorter chain polysulfides. This background should not be disregarded and has to be subtracted before attempting any peak fitting.

We thank the reviewer for the comments and the observations. We note that in our potentiostatic experiments, we employed an intermediate Li_2S_4 electrolyte, rather than the long-chain Li_2S_8 or Li_2S_6 polysulfides used in previous studies. While we appreciate the reviewer's point and concerns, we believe that background subtraction is not needed in our work for the reasons listed below. 1. For solutions of long-chain polysulfides such as Li_2S_8 and Li_2S_6 , the main purpose of background subtraction is to remove the contributions from their reduction processes to Li_2S_4 . By directly using a Li_2S_4 electrolyte as the active material we essentially preclude such contributions. 2. In previous studies, people have compared the capacities among different cathode materials, at the same applied potential. However, we did experiments under different overpotentials, which will clearly affect the dynamic equilibria of the polysulfide interconversions (disproportionation and conproportionation reactions) in solution. For example, at lower overpotentials, there will be a higher fraction of the more oxidized polysulfides. We can expect that higher backgrounds will be subtracted from them, leading to bigger differences when comparing the capacities. Supplementary Figure 17 presents the results after background subtraction as references, which is consistent with our hypotheses. It is evident that this treatment will affect the actual amounts/concentrations of the active materials participating in the polysulfide redox processes, which will inevitably introduce new variables that can decrease the accuracy of the measurements and hence, the conclusions that can be drawn. Taking these into consideration, we employed the original data for in-depth analysis and provided the subtracted results as references in Supplementary Figure 17. Additional comparisons, analysis, and explanations have been incorporated into the revised version of the manuscript (Page 15) and SI (Pages 18-19).

Supplementary Figure 17. Current-time transients of Li₂S deposition at different potentials (background subtraction). Current-time transients of Li₂S nucleation and growth obtained at (a) 2.0 V, (b) 2.05 V and (c) 2.1 V (vs. Li⁺/Li). The reduction processes of long-chain polysulfides were subtracted from the total capacity by two exponential functions. (d) Corresponding discharge capacities after background subtraction.

c. Specific proof of discrepancies between the data – for the reduction of polysulfides at 2.05 V vs. Li/Li+, you determined 2D deposition. This means that only a very thin (a quasi monolayer) of Li₂S is supposed to form on the electrode surface and that its further formation is limited due to the nonconductive deposit passivating the surface. Such a deposit can only be a few nm thick by default. Yet the SEM data shows very thick (on the order of μm) deposits on the carbon fibers.

We thank the reviewer for the comments. We need to keep in mind that Li₂S is electronically insulating but ionically conducting. Multilayer Li₂S electrodeposition can be achieved by the polysulfide reduction at the three-phase boundary of the carbon substrate, Li₂S, and the electrolyte. The continuous nucleation and growth processes of Li₂S on multi-sites of the electrode enabled the formation of Li₂S with larger sizes (μm thick). In addition to numerous publications using SEM (Yuegang Zhang et al. *Adv. Energy Mater.*, **2015**, *5*, 1501369; Jun Liu et al. *Nature Energy*, **2017**, *2*, 813-820, etc.), there is also a significant number of reports employing imaging characterizations such as, *in situ* AFM, SECM and TEM (Rui Wen and Li-Jun Wan et al. *Angew.Chem. Int.Ed.*, **2017**, *56*, 14433–14437, *J. Am. Chem. Soc.*, **2018**, *140*, 8147–8155; Leela Mohana Reddy Arava et al. *Nano Lett.*, **2019**, *19*, 5229–5236; Xingcheng Xiao et al. *Adv. Energy Mater.* **2015**, *5*, 1401752, etc.) that observed the deposited Li₂S at micron and submicron scales. We have added these comments and discussions to the revised manuscript (Pages 16-17).

7. Your comment on whether the mechanism of operation is through a stepwise mechanism or through parallel pathway is not convincing. Partly because the attribution of bands to different polysulfide species is questionable (points 3. and 4.) and partly because it is barely discussed. Does this mean that the authors doubt the presence of disproportionation reactions? The conclusion on the stepwise versus parallel pathway could not be made with the data presented accounting to the short-comings of the techniques/methods employed.

We thank the reviewer for the comments. As we articulated above (see the responses to questions 3 and 4), our attributions of the polysulfides were based on very systematic investigations combining *ex situ*

experiments in the liquid and/or frozen states, *in situ* measurements in Li-S cells, calculations, and proposed reaction mechanisms. (J. P. Lelieur et al. *Inorganic Chemistry*, **1988**, *27*, 73-80; M. Hagen et al. *Journal of The Electrochemical Society*, **2013**, *160*, A1205-A1214; Qiang Zhang et al. *Small Methods*, **2017**, *1*, 1700134; K. Zaghbi et al. *Sustainable Energy Fuels*, **2017**, *1*, 737). As discussed in the manuscript, in Li_2S_4 solutions, we can observe the peak at 400 cm^{-1} that represents long chain polysulfides that arise as a result of the close Gibbs free energies and disproportionation reactions among the different polysulfides. The rapid decrease of this peak during potentiostatic reactions at 2.0 V indicates the faster reduction rates of the long chain polysulfides at a relatively larger overpotential, which, in turn, reflects the accuracy of our identification and detection (Figure 2 and Supplementary Figure 13).

It is worth emphasizing that the stepwise reduction process that we present in the manuscript does not claim that no other polysulfides exist in solution when one forms. We proposed the stepwise and parallel evolution mechanisms based on the changing trends of different polysulfides as characterized by the relative intensities and their changes. During the discharge processes, non-synchronous changes in the intensities of the long-chain and shorter chain polysulfides were observed, providing clear evidence that the transformations from long-chain are essential to the formation of shorter ones. The prominent presence/function of the intermediate products reveals the characteristics of the stepwise reaction. This is clearly different from their simultaneous changes during the recharge process, directly distinguishing the stepwise discharge mechanism from the parallel recharge one. These phenomena can be due, at least in part, to the slower diffusion processes of the longer chains and the sluggish solid-liquid transformations relative to the liquid-liquid ones, revealing important information of Li-S redox mechanisms and inspiring strategies for effectively catalyzing these processes. Additional explanations and discussions have been added to the revised version of the manuscript (Pages 7 and 22-23, 26) and SI (Page 25).

8. I would wish for more discussion regarding the in-depth mapping of the cross-sectional distribution of polysulfides. On the image it seems that the polysulfide signal disappears even before reaching the anode. Is this a consequence of dilution? Can this be used to determine disproportionation reactions between polysulfides?

We thank the reviewer for the comments and suggestions. The color contrasts of all images in our manuscript remain consistent enabling clear quantification. For the in-depth mapping, it is reasonable that the signals of the polysulfides are less intense at the edges than in the center regions of the cathode since most of the sulfur clusters are distributed in the middle of the cathode. As shown in Figure 5i, the front edges of the polysulfide gradually stretch into the electrolyte, driven by the concentration gradient, which further decreases the signals at the edges. The disappearance/fading of these signals can be related to various reasons, including their further dilution, disproportionation reactions, and diffusion to the regions that were not under observation. In future investigations, we will focus our efforts on the disproportionation and comproportionation reactions between polysulfides. We have added these comments and discussions to the revised manuscript (Page 23).

Reviewer #4 (Remarks to the Author):

The work presented in this manuscript explores the use of operando confocal Raman microscopy to gain mechanistic insight into polysulfide formation kinetics. As evident from this piece of work and previous reports exploring the use of Raman spectroscopy or other operando/in-situ techniques to understand the mechanism and kinetics of lithium-sulphur redox, it is clear that unambiguous elucidation of the discrete mechanistic steps is extremely challenging. The reduction reactions of sulphur, including its transformation to long-chain polysulfides and intermediate-chain polysulfides, have been discussed in many theoretical and experimental articles already. The novelty of this work is indeed the use of Raman microscopy in this context, and this work further supports the existing idea of polysulfide formation and

reaction kinetics (arguably first order as suggested in some previous reports as well). In fact, the use of other microscopy techniques (in-operando transmission X-ray microscopy, in-situ X-ray fluorescence microscopy, electron microscopy, etc) in this context have also been reported. While this study nicely demonstrates the use of Raman microscopy and provides additional evidence to the pool of information available on this topic in the literature, I believe there's no notable new insight or an unambiguous elucidation of the reaction mechanism based on this new approach that would make it suitable for Nature Communications. The use of operando confocal microscopy in this context is interesting, but this manuscript is more suitable for a more specialized journal in my opinion.

While we appreciate the reviewer's comments, we respectfully disagree with his/her final assessment. It is indeed extremely challenging to achieve an unambiguous elucidation of the Li-S redox mechanism and kinetics. Even though numerous theoretical and experimental efforts have been devoted to these investigations, fundamental insights into many crucial issues of Li-S redox have not been established, including the reaction order of different sub-steps, reaction rates under various environments, the nucleation and growth processes of Li_2S , and transformation mechanism and kinetics between/among different polysulfides. This is partly because of the complexity of the multistep Li-S redox reactions, involving solid-liquid-solid phase transformations and the diffusion of soluble polysulfides. In addition, it is quite difficult to simultaneously track, identify, and quantify the different (poly)sulfide species in real-time during the discharge and charge processes. For most of the advanced characterization techniques, including but not limited to the ones the reviewer mentioned (*in operando* transmission X-ray microscopy, in-situ X-ray fluorescence microscopy, electron microscopy, etc.), the limited resolution in spatial distribution and/or the lack of clear distinction of different polysulfides restrict the accurate measurement and comprehensive diagnosis of the reaction mechanism and kinetics of Li-S redox (Michael F. Toney et al. *J. Am. Chem. Soc.*, **2012**, *134*, 6337–6343; Xiao-Qing Yang et al. *Adv. Energy Mater.*, **2015**, *5*, 1500072; Xingcheng Xiao et al. *Adv. Energy Mater.* **2015**, *5*, 1401752, etc.). Providing such insights and achieving the level of quantitative understanding that our work did, is a unique and important accomplishment in the Li-S field, which can extend our knowledge of Li-S redox processes, inspire novel ideas and thinking that can yield significant advances in the field of Li-S batteries. It can also stimulate further exploration of Raman imaging to investigate multiple complex reaction processes. Therefore, we respectfully disagree with the reviewer's assessments and recommendation. Below we explain and highlight the significance and novelties of our work in detail, mainly focusing on the exploration and application of the fast and high-resolution Raman imaging, the systematic elucidation of the reaction kinetics of sulfur and polysulfides, and the clear identification and visualization of different polysulfides, elucidating their transformation mechanisms.

1. A number of reviews have summarized the *in situ/operando* investigations of Li-S reactions using a variety of advanced characterization techniques, including synchrotron-based methods, electron microscopy, optical microscopy, and various spectroscopic methods (Liqiang Mai et al. *Nanoscale*, **2017**, *9*, 19001; Hong Li et al. *Adv. Funct. Mater.*, **2018**, *28*, 1707543, etc.). They all illustrate specific applications for Li-S research. For example, in-operando transmission X-ray microscopy can visualize the evolution of sulfur (Nae-Lih Wu et al. *Journal of Power Sources*, **2014**, *263*, 98e103). UV-vis can be employed to distinguish between different polysulfides and elucidate the reaction pathway (Yi-Chun Lu et al. *J. Phys. Chem. Lett.*, **2016**, *7*, 1518–1525). However, a thorough understanding of the complex Li-S redox requires the combination of high-resolution imaging, simultaneous identification of different (poly)sulfide species, rapid and multi-site detection, and quantitative analysis, which, to date, has been very difficult to achieve by these imaging and spectroscopic measurements. The confocal Raman microscopy approach that we employed in our work enables simultaneous identification of sulfur and various polysulfides, and imaging the exact focal plane at high resolution, providing compelling evidence for many important questions regarding the Li-S mechanisms. In our work, during both potentiostatic and galvanostatic redox processes, we have successfully monitored the spatial distribution and evolution of sulfur and different length

polysulfides in real-time. The diffusion processes of different polysulfides were clearly captured by cross-sectional mapping from the cathode surface into the electrolyte. The excellent performance of *operando* confocal Raman microscopy in terms of speed (150×150 μm² captured every 5 mins), identification of different sulfide species, and spatial resolution, down to submicron level, brings new opportunities for us to investigate and characterize the Li-S redox processes in a comprehensive and quantitative manner.

2. It is important to note that, in spite of numerous studies, the fundamental understanding of the Li-S redox mechanism and kinetics, including the first-order reaction kinetics and potential-dependent reaction rates of sulfur and polysulfides, remain unresolved. In the Li-S field, many previous reports have focused on comparing the reaction pathways among modified systems, like host optimized/interlayer added cathode materials, solvent/salt/additive mediated electrolytes, and SEI improved anodes (Quan-Hong Yang et al. *Energy Environ. Sci.*, **2017**, *10*, 1694; Jang Wook Choi et al. *Adv. Energy Mater.*, **2020**, 2001456; Liqiang Mai et al. *Nanoscale*, **2017**, *9*, 19001, etc.). Few investigations have focused on and dedicated to the most fundamental understanding such as the reaction order, various factors of and affecting reaction rates, and the kinetics of S/Li₂S electrodeposition. In terms of reaction kinetics, Wu et al. directly applied first-order kinetics to fit the sulfur reduction process (Gewirth et al. *ACS Appl. Mater. Interfaces*, **2015**, *7*, 1709). Their simple treatment of the multi-step redox processes using high overpotentials, and the lack of analysis focusing on the reaction of polysulfides, led to the conclusion that short-chain S₃²⁻ is directly formed from sulfur decomposition. This is inconsistent with the generally accepted views about the solid-liquid-solid transformations and the initial formation of long-chain polysulfides during sulfur reduction. In contrast, we have demonstrated, in compelling fashion, the first-order kinetics and calculated the potential-dependent reaction rates of sulfur reduction and polysulfide redox, based on accurate visualization, quantification, and comparison among various models. The conductivity dependency of sulfur reduction and the concentration dependency of polysulfides have also been characterized. New insights into the correlations between/among the overpotential, the shape of the i-t curve, capacity, and Li₂S morphologies have been gained, which have, in turn, provided very important information on Li₂S nucleation and growth that can guide future catalyst design. The step-by-step measurement and systematic elucidation, articulated in our work, have provided fundamental insights of Li-S redox mechanisms and kinetics that have not been previously presented.

3. We have identified, visualized, and investigated, in detail, the transformations between/among different polysulfides during discharge and charge processes. The in-depth understanding of the polysulfide transformations is tremendously important but extremely challenging because of the inherent difficulties in imaging and distinguishing between/among different polysulfides. In addition, spectroscopic methods are usually not good at multi-site detection for accurate quantification. In our work, polysulfides with different lengths were identified and individually visualized during the discharge and charge processes. The Raman images captured at the cathode surface and deep into the electrolyte clearly illustrate their spatial distribution and differences in intensity changes at different states of discharge/charge, showing a stepwise reduction process but a parallel oxidation mechanism during cell operation. These observations provide compelling evidence for understanding the solid-liquid-solid transformation processes during Li-S redox. The differences between the discharge and charge processes can be due, at least in part, to the slower diffusion processes of the longer chains and the sluggish solid-liquid transformations relative to the liquid-liquid ones. Our findings have enhanced and expanded our understanding of polysulfide redox and we believe will facilitate catalyst design in the Li-S field.

Therefore, we strongly feel that our present work represents a significant advance in our understanding of the Li-S redox mechanism and kinetics and a demonstration of the power of Raman imaging for Li-S research, providing compelling evidence, systematic analysis, and valuable new and important insights

regarding Li-S electrochemistry and Raman imaging exploration. We feel that it is one of the few studies that have enable a systematical elucidation and gain the most fundamental understanding of the Li-S redox mechanism and kinetics. We have incorporated these comments into the revised version of the manuscript (Pages 3-4, 15 and 23).

REVIEWER COMMENTS

Reviewer #2 (Remarks to the Author):

The authors have done a very thorough job of answering the inquiries of the reviewers. Because of the wide variety of research going on in Li-S, it's difficult to develop a study for Li-S that merits publication in a journal like Nature Communications. However, I think the fundamental nature of the work shown here and the mechanistic insight that is gained is worthy of such a journal. It is clear from the reviewer comments that the authors have certainly put forth a discourse worthy of discussion, and they have defended it well. The science is sound and the study is thorough. I also believe the paper has been improved through the peer review. I suggest that this paper be considered for publication.

Reviewer #3 (Remarks to the Author):

Reviewer's comments:

In the revised manuscript entitled "Mechanism and Kinetics of Lithium-Sulfur Redox via Operando Confocal Raman Microscopy", the authors changed some discussion according to the reviewer's comments. Nevertheless, I find that the concerns, questions and comments I had about the work were not sufficiently taken into consideration.

1. Regarding the provided proposed reactions in the revised Supplementary Table 2: non-radical polysulfide species with odd number of sulfur atoms were completely disregarded? If this paper aims to elucidate the underlying mechanism of polysulfide generation/evolution you cannot dismiss half the species present.
2. The Raman peak analysis for different polysulfide chain lengths is not as straightforward as the authors claim. As stated in the rebuttal letter, two peaks are present in the 400 - 450 cm^{-1} region in polysulfide solutions, with the difference being that one of the peaks is higher for polysulfides with longer chain lengths (since it corresponds to central S-S bond vibrations) and the other is higher for

polysulfides with shorter chain lengths (since it corresponds to terminal S-S bond vibrations). This cannot be simplified to attributing one peak to long-chain polysulfides and the other to the short-chain polysulfides, even if some literature reports have already done this. Both peaks are present for different chain lengths of polysulfides, yet the ratio between them is different depending on exact length. If the authors want to claim that one peak is for longer chain polysulfides and one is for shorter chain polysulfides, then please provide sufficient experimental proof of this.

3. I have suggested that the analysis of potentiostatic current response for determination of Li₂S deposition is wrong due to several reasons. I still remain convinced that the analysis provided is wrong:

a. In the letter, the authors claimed that they have carefully investigated the classical theories analyzing the kinetics of nucleation and growth". If so, please explain the following: the equations presume diffusion limited process and a constant effective diffusional coefficient and bulk concentration. Please provide evidence that your experiments are done in such a way that these conditions are met.

b. Even if we forget about the peak equations, the background subtraction in the analysis is vital. The current decrease before peak occurrence cannot be solely from double layer charging. See for example the black curve on Figure 3b – what evidence do you have that for the first 30 min, the sole process providing the current response and approx. 20 % of total final capacity obtained from the cell (I am estimating by eye) is double layer charging? This simply cannot be true for the battery type cell as you employed. Although you provided some options of background subtraction, these are not the only ways of how this could be done. Before claiming that this analysis is true, please provide experimental proof why the background cannot be higher (closer to the peak, this also importantly changes the peak shape/fitting/mechanism determinations). Also, stating that you circumvented this issue by using Li₂S₄ solution, which supposedly directly reduces solely to Li₂S shows limited understanding of polysulfide equilibria or that you completely disregard the possibility of formation of Li₂S₃ or Li₂S₂ (see also my point 1).

4. I pointed out a discrepancy between the fitted data (showing 2D deposition, i.e. very thin deposited layer, nm size) and SEM data (showing micrometer size depositions). The authors responded that multilayer Li₂S deposition can be achieved through polysulfide reduction at the three-phase boundary of the carbon, Li₂S and electrolyte because Li₂S is electronically insulating, yet it conducts ions. This is confusing – you have fitted data to a 2D deposition (which I am claiming is wrong, see point 3), yet now you are saying that this is multilayer deposition. How can this coincide? Also, how do you propose ionic conductivity should help with this? The charge transfer reaction can only take place at the electrode surface (because Li₂S is an electronic insulator), we can get Li⁺ ions there too (because Li₂S have rather good Li⁺ conductivity). But how can we also get polysulfide ions there in order to reduce them? Again, sufficient proof for the conclusions has not been provided.

Point-by-point response

Below we include the reviewers' comments in Black, and our responses (in Blue).

Reviewer #2 (Remarks to the Author):

The authors have done a very thorough job of answering the inquiries of the reviewers. Because of the wide variety of research going on in Li-S, it's difficult to develop a study for Li-S that merits publication in a journal like Nature Communications. However, I think the fundamental nature of the work shown here and the mechanistic insight that is gained is worthy of such a journal. It is clear from the reviewer comments that the authors have certainly put forth a discourse worthy of discussion, and they have defended it well. The science is sound and the study is thorough. I also believe the paper has been improved through the peer review. I suggest that this paper be considered for publication.

We sincerely appreciate the reviewer for the positive comments, recommendation and for recognizing the importance of the work and our efforts to contribute mechanistic insights to this important field.

Reviewer #3 (Remarks to the Author):

In the revised manuscript entitled "Mechanism and Kinetics of Lithium-Sulfur Redox via Operando Confocal Raman Microscopy", the authors changed some discussion according to the reviewer's comments. Nevertheless, I find that the concerns, questions and comments I had about the work were not sufficiently taken into consideration.

1. Regarding the provided proposed reactions in the revised Supplementary Table 2: non-radical polysulfide species with odd number of sulfur atoms were completely disregarded? If this paper aims to elucidate the underlying mechanism of polysulfide generation/evolution you cannot dismiss half the species present.

We thank the reviewer for the comments and have added reactions of non-radical polysulfides with odd number of sulfur atoms in the revised Supplementary Table 2.

Supplementary Table 2. Main sub-steps of the sulfur reduction process and accompanying disproportionation and conproportionation reactions.

Stages	Proposed reactions
Solid-liquid reduction of sulfur to polysulfides	$S_8 + 2e^- \rightarrow S_8^{2-}$
Polysulfide transformation	$S_8^{2-} + 2e^- \rightarrow 2S_4^{2-}$ $S_6^{2-} + e^- \rightarrow 3/2S_4^{2-}$ $S_7^{2-} + 2e^- \rightarrow S_4^{2-} + S_3^{2-}$ $S_5^{2-} + 2e^- \rightarrow S_3^{2-} + S_2^{2-}$
Formation of short-chain sulfides	$S_4^{2-} + 2e^- \rightarrow 2S_2^{2-}$ $S_3^{2-} + 2e^- \rightarrow S^{2-} + S_2^{2-}$ $S_2^{2-} + 2e^- \rightarrow 2S^{2-}$
Disproportionation and conproportionation reactions	$S_8^{2-} \rightarrow 1/4S_8 + S_6^{2-}$ $S_6^{2-} \rightarrow 2S_3^{*}$ $2S_6^{2-} \rightarrow S_5^{2-} + S_7^{2-}$ $S_4^{2-} \rightarrow 3/5S_6^{2-} + 2/5S^{2-}$ $2S_4^{2-} \rightarrow S_5^{2-} + S_3^{2-}$ $2S_3^{2-} \rightarrow S^{2-} + S_5^{2-}$ $3/5S_6^{2-} + 1/5S_2^{2-} \rightarrow S_4^{2-}$

2. The Raman peak analysis for different polysulfide chain lengths is not as straightforward as the authors claim. As stated in the rebuttal letter, two peaks are present in the 400 - 450 cm^{-1} region in polysulfide solutions, with the difference being that one of the peaks is higher for polysulfides with longer chain lengths (since it corresponds to central S-S bond vibrations) and the other is higher for polysulfides with shorter chain lengths (since it corresponds to terminal S-S bond vibrations). This cannot be simplified to attributing one peak to long-chain polysulfides and the other to the short-chain polysulfides, even if some literature reports have already done this. Both peaks are present for different chain lengths of polysulfides, yet the ratio between them is different depending on exact length. If the authors want to claim that one peak is for longer chain polysulfides and one is for shorter chain polysulfides, then please provide sufficient experimental proof of this.

We thank the reviewer for the comments. It is worth re-emphasizing that polysulfides with different lengths co-exist in solution. As the reviewer is no doubt aware of, there will always be a distribution of different chain length polysulfides during cell operation, which presents experimental challenges for characterization. Given this constraint, what one can do from an experimental point of view, is provide

Figure 5. Operando confocal Raman investigation of Li-S redox processes during the galvanostatic discharge process. (a) Voltage profile, operando Raman (b) spectra and (c-j) mapping results of sulfur and polysulfides during discharge at a rate of 0.02C ($1\text{C} = 1672 \text{mAh g}^{-1}$). The red, yellow, and blue colors in mapping images represent sulfur, polysulfide S_x^{2-} , $x=6-8$, and 3-5, respectively. The color contrasts remained consistent for comparison. Different DODs are indicated with %. (d) presents the area changes of the pristine sulfur clusters at different DODs. (f) On surface and (i) in depth mapping images of the polysulfide evolution during the discharge process. (g) and (j), respectively, provide the quantified data on surface and in depth, extracted from (f) and (i). The orange and blue colors represent long-chain and intermediate polysulfides, respectively. In the right of (i), for clear visualization, color mapping of cross-sectional polysulfide evolution was extracted from the Raman mapping images in the left. (h) In depth mapping that reveals the cross-sectional distribution of the polysulfides at 5% DOD.

As the reviewer is no doubt aware of, there will always be a distribution of different chain length polysulfides during cell operation, which presents experimental challenges for characterization. Given this constraint, what one can do from an experimental point of view, is provide

plausible mechanistic pathways that are supported by experimental observation. That is, in fact, what we have done in this work. The galvanostatic and potentiostatic experimental results have been provided to show that the two peaks at 400 and 450 cm^{-1} can be employed to reveal/reflect the evolution of long-chain and intermediate polysulfides, respectively. As shown in Figure 5, the *operando* spectra, on surface and in depth mapping images quantitatively revealed the stepwise discharge process of the long-chain and intermediate polysulfides, providing compelling evidence of the differences of the polysulfides' evolution. For the potentiostatic reduction reaction, in the *operando* spectra and mapping data (Figure 2 and Supplementary Figure 13), the decreases in the peak region around 400 cm^{-1} are evidently faster than those around 450 cm^{-1} , indicating the higher reduction rate of the longer-chain polysulfides relative to the shorter chain polysulfides at the same potential. We have additionally modified our description in the revised manuscript in an effort to provide a clearer statement. "The peaks at 405 cm^{-1} and 453 cm^{-1} were employed to illustrate the evolution of the long-chain Li_2S_x ($x=6-8$) and intermediate-chain Li_2S_x ($x=3-5$) polysulfides, respectively, during potentiostatic and galvanostatic experiments."

Figure 2. Reduction of polysulfides at 2.0 V vs. Li⁺/Li. (a) Chronoamperometric current-time transient of polysulfide reduction at 2.0 V (vs. Li⁺/Li). *Operando* Raman (b) spectra and (c-f) mapping images of the cathode during reduction in 1.0 M Li_2S_4 electrolyte. The blue color in (c-f) represents the region at 400 cm^{-1} . The color contrast remains consistent for quantification. (g) Plots of the concentration changes in 1.0, 0.75, and 0.5 M Li_2S_4 electrolytes with time. (h) Linear fitting of the logarithm of Li_2S_4 concentration evolution with time according to first-order kinetics, $R^2=0.982$, 0.979 , 0.968 for 1.0, 0.75, and 0.5 M electrolytes, respectively. (i) Values of k_{ps} extracted from the fits.

Supplementary Figure 13. Reduction of long-chain polysulfides at 2.0 V vs. Li⁺/Li. *Operando* Raman (a) spectra and (b-e) mapping images during the reduction process. The yellow color in (b-e) shows the signal from long-chain S_x²⁻, x=6-8 polysulfides, centered at 400 cm⁻¹ with a band width of 20 cm⁻¹. The color contrast is consistent for quantification, as shown under (e). (f) Raw data of the area changes of long-chain polysulfides during the reduction process.

3. I have suggested that the analysis of potentiostatic current response for determination of Li₂S deposition is wrong due to several reasons. I still remain convinced that the analysis provided is wrong: a. In the letter, the authors claimed that they have carefully investigated the classical theories analyzing the kinetics of nucleation and growth". If so, please explain the following: the equations presume diffusion limited process and a constant effective diffusional coefficient and bulk concentration. Please provide evidence that your experiments are done in such a way that these conditions are met.

We thank the reviewer for the comments. In an effort to address the issue of diffusion coefficients, we carried CV and GITT experiments for the reduction of polysulfides. In the first case we carried out CV experiments, using Li₂S₄ as the catholyte, at different scan rates (Supplementary Figure 18a). A plot of peak current vs. the square root of the scan rate was linear

Supplementary Figure 18. Diffusion-controlled process of polysulfide reduction. (a) CV profiles of a Li – polysulfide battery using Li₂S₄ as the catholyte at various scan rates from 0.1 mV s⁻¹ to 0.5 mV s⁻¹. (b) The plot of the polysulfide reduction peak current vs. square root of scan rates.

(Supplementary Figure 18b), indicating that the process was diffusion controlled. Using the Randles–Sevcik equation:

$$I_p = 2.69 \times 10^5 n^{3/2} A D^{1/2} C v^{1/2} \quad (1)$$

where I_p is the peak current, n is the charge transfer number, A is the geometric area of the active electrode, D is the lithium-ion diffusion coefficient, C is the concentration of Li^+ , and v is the potential scan rate, we calculated the diffusion coefficient to be $7.47 \times 10^{-9} \text{ cm}^2 \text{ s}^{-1}$. In addition, we employed GITT to measure the diffusion coefficients of the polysulfides during the discharge process (at different DOD values). The analysis yielded values of the order of $10^{-9} \text{ cm}^2 \text{ s}^{-1}$ (Supplementary Figures 8b-c). In an effort to establish the constancy of the polysulfide concentration, we have employed the signals of polysulfides for extended time periods using *operando* Raman mapping (Supplementary Figure 10). The signals remained stable (Supplementary Figure 10f), indicating a constant bulk concentration. We have moved the section “Nucleation and growth mechanism of Li_2S ” to the SI and incorporated the above discussion into the revised manuscript and SI.

Supplementary Figure 8. (b) Raw data of GITT using Li_2S_4 as the catholyte during the first discharge process. The shallow slope before the plateau could be due to the contribution from the co-existing long-chain polysulfides. (c) Diffusion coefficients calculated from GITT.

Supplementary Figure 10. *Operando* confocal Raman investigation of Li – polysulfide cells at open circuit.

(a) Plot of the potential changes at open circuit. *Operando* Raman (b) spectra and (c-e) mapping images during cell storage. The blue color in (c-e) shows the signal from intermediate polysulfides, centered at 453 cm^{-1} with a band width of 20 cm^{-1} . The color contrast is consistent in images and shown under (e). (f) Raw data of the area changes of the mapping results during cell storage.

b. Even if we forget about the peak equations, the background subtraction in the analysis is vital. The current decrease before peak occurrence cannot be solely from double layer charging. See for example the black curve on Figure 3b – what evidence do you have that for the first 30 min, the sole process providing the current response and approx. 20 % of total final capacity obtained from the cell (I am estimating by eye) is double layer charging? This simply cannot be true for the battery type cell as you employed. Although you provided some options of background subtraction, these are not the only ways of how this could be done. Before claiming that this analysis is true, please provide experimental proof why the background cannot be higher (closer to the peak, this also importantly changes the peak shape/fitting/mechanism determinations).

Also, stating that you circumvented this issue by using Li2S4 solution, which supposedly directly reduces solely to Li2S shows limited understanding of polysulfide equilibria or that you completely disregard the possibility of formation of Li2S3 or Li2S2 (see also my point 1).

We thank the reviewer for the comments. While there will be some double layer contributions, we agree with the reviewer that they cannot be solely responsible for the charge/current prior to the peak. The “incubation” time/period of the current decrease prior to the peak could also be attributed, at least in part, to the reduction of higher-order polysulfides. In the revised manuscript, we have modified our description as follows. “In all three cases, the current-time transients exhibited an initial high current and decrease prior to the emergence of a peak, which could be due to relaxation processes along with the reduction of higher-order polysulfides.”

A critical value of the overpotential is necessary for the nucleation of Li₂S. As shown in Supplementary

Supplementary Figure 17. Current-time transients of Li₂S deposition at different potentials (background subtraction). (a) Current-time transients performed at 2.12 V. Background subtraction using two exponential functions at (b) 2.12 V, (c) 2.1 V, (d) 2.05 V and (e) 2.0 V (vs. Li⁺/Li). (f) Corresponding discharge capacities after background subtraction. (g-i) Comparison of the original curves and the ones after subtraction and (j-l) the fitting curves at 2.1, 2.05, and 2.0 V.

Figure 17a, the I-t curve performed at 2.12 V exhibits a continuous decrease of the current and can be fit as the superposition of two exponential functions, which we treat as the reduction of high-order polysulfides (Supplementary Figure 17b). A sum of two exponential functions was employed as a fit for the background for the I-t curves performed at higher overpotentials. In addition, as would be expected, the ratio of Li₂S nucleation in the total capacity would increase, at least not decrease, in cases at higher overpotentials. We performed the background subtraction based on the above method and described rules (Supplementary Figure 17c-e). The method of background subtraction and its application in Li-S studies have been employed in earlier literature reports (Yet-Ming Chiang et al. *Adv.Mater.*, **2015**, *27*, 5203–5209).

The peak fitting and mechanistic insights based on the use of BFT and SH models are more related to the curve shape around I_{\max} and the required time to reach the I_{\max} , t_{\max} . We have further modified the curves with the background subtraction. The curves after subtraction and the corresponding fitting curves are presented in Supplementary Figure 17g-l. While the values of $(I/I_{\max})^2$ around the peak exhibited some slight fluctuations, after background subtraction, the trends of the nucleation models, from 2D to 3D with increasing overpotential, were very similar to the results obtained using the original data.

We agree that Li₂S₃ and/or Li₂S₂ will inevitably form during the reduction of Li₂S₄. However, it would be extremely difficult to prepare Li₂S₃ and Li₂S₂ even if we mixed stoichiometric amounts of Li₂S and S, due to the disproportionation and conproportionation reactions. The complex interconversion and interplay of the different polysulfides, as well as potential short-chain sulfide deposits, would affect the accuracy of our detection. Thus, the Li₂S₄ solution was used for the investigation of polysulfide reduction. We have expanded our explanation and moved this “Nucleation and growth mechanism of Li₂S” section to SI in the revised version of the manuscript.

4. I pointed out a discrepancy between the fitted data (showing 2D deposition, i.e. very thin deposited layer, nm size) and SEM data (showing micrometer size depositions). The authors responded that multilayer Li₂S deposition can be achieved through polysulfide reduction at the three-phase boundary of the carbon, Li₂S and electrolyte because Li₂S is electronically insulating, yet it conducts ions. This is confusing – you have fitted data to a 2D deposition (which I am claiming is wrong, see point 3), yet now you are saying that this is multilayer deposition. How can this coincide? Also, how do you propose ionic conductivity should help with this? The charge transfer reaction can only take place at the electrode surface (because Li₂S is an electronic insulator), we can get Li⁺ ions there too (because Li₂S have rather good Li⁺ conductivity). But how can we also get polysulfide ions there in order to reduce them? Again, sufficient proof for the conclusions has not been provided.

We thank the reviewer for the comments and apologize for the confusion. From the SEM images, we can observe that the morphologies of Li₂S deposits change from the films to accumulated islands with increasing overpotentials, which can be related to the higher values of peak current at higher overpotentials and leads to the higher capacity. In the revised manuscript, we have modified our description and moved the “Nucleation and growth mechanism of Li₂S” section to SI.